



# Meteorological export and deposition fluxes of Black Carbon on glaciers of the central Chilean Andes

Rémy Lapere[1], Nicolás Huneeus[2], Sylvain Mailler[1,3], Laurent Menut[1], and Florian Couvidat[4]

[1]Laboratoire de Météorologie Dynamique, IPSL, École Polytechnique, Institut Polytechnique de Paris, ENS, Université PSL, Sorbonne Université, CNRS, Palaiseau, France
[2]Department of Geophysics, Universidad de Chile, Santiago, Chile
[3]École des Ponts, Université Paris-Est, 77455 Champs-sur-Marne, France
[4]Institut National de l'Environnement Industriel et des Risques, Verneuil-en-Halatte, France

**Correspondence:** Rémy Lapere (remy.lapere@lmd.ipsl.fr)

**Abstract.** Air pollution in the central zone of Chile not only is a public health concern, but also threatens water resources and climate, in connection with the transport and deposition of black carbon (BC) from urban centers onto the glaciers of the Andes Cordillera. Chemistry-transport simulations reveal a seasonal dichotomy in the flux and latitudinal pattern of BC deposition on glaciers of the central Chilean Andes. The average deposition flux of BC on glaciers between 30°S and 37°S is

4 times larger in winter, affecting mostly low elevation glaciers, whereas the smaller summertime flux affects glaciers evenly, irrespective of their elevation. The contribution of emissions from Santiago city is dominant in summertime with more than 50% along the Andes, but minor in wintertime with less than 20% even close to the capital city. Transport at larger scales and more local sources likely account for the remaining flux. The superimposition of synoptic-scale circulation and local mountain-valley circulation along the Andes cordillera drives the differences between summertime and wintertime deposition

fluxes and generates a greater meteorological export potential during summer months. Future emissions and climate projections suggest that under the RCP8.5 scenario the gap between summertime and wintertime BC export and deposition flux could decrease, thereby pointing to summertime emission control gaining relevance. The chemistry-transport modeling approach for BC deposition on the Andes sheds light on the importance of the often disregarded summertime emissions on the radiative balance of its glaciers, particularly in the vicinity of Santiago.

## 1 Introduction

Urban areas in the central region of Chile (extending roughly between 30 and 37°S) face concerning air pollution issues, especially regarding fine particulate matter ($PM_{2.5}$) (e.g. Saide et al., 2016; Barraza et al., 2017; Lapere et al., 2021b). Strong anthropogenic emissions of primary particulate matter and precursors mostly stem from dense road traffic and industrial activities all year round, and a large additional contribution from wood burning for residential heating in wintertime, particularly in

the metropolitan region of Santiago (Barraza et al., 2017; Marín et al., 2017; Mazzeo et al., 2018; Álamos et al., 2022). Among these emissions, black carbon (BC) particles constitute a large fraction, in relation with incomplete combustion of fossil fuel and biomass (Shrestha et al., 2010; Villalobos et al., 2015).



Besides the health issues associated with air pollution (Ilabaca et al., 1999), light-absorbing particles such as BC have a strong radiative impact once deposited on snow and ice, inducing a reduction in their albedo, hence accelerating their melting (e.g. Kang et al., 2020). BC was found in significant amounts in snow of the central Chilean Andes, creating an estimated additional radiative forcing of $1.4\,\mathrm{W\,m^{-2}}$ in the region of Santiago for winter months (Rowe et al., 2019), which can go up to $10\,\mathrm{W\,m^{-2}}$ for the most impacted sites (Cordero et al., 2022). Dust particles emitted by mining activities in the Andes are also found to affect ice and snow albedo (Barandun et al., 2022).

The region has been facing a hydrological deficit since the last decade in relation with the recent mega-drought (Garreaud et al., 2017) so that the additional accelerated melting of glaciers and snowpack triggered by impurities leads to an even higher stress on water resources at the end of summer. In a short-term perspective an accelerated melting increases river runoff and therefore freshwater availability. On the long-term however, a peak water is reached and river runoff starts decreasing and eventually stops, when glaciers have sufficiently retreated (Hock et al., 2019). Ayala et al. (2020) suggests that glaciers of central Chile may have already passed peak water, making accelerated melting even more concerning. In this context it is paramount to expand the knowledge on BC transport towards, and deposition on, glaciers of the Chilean Andes.

Central Chile is a narrow strip of land surrounded by the Pacific Ocean in the West and the Andes Cordillera in the East. As a result, the atmospheric circulation is driven both by synoptic-scale conditions related to the position of the South Pacific High (SPH) over the Pacific Ocean, and by local mountain-valley circulation dynamics. Mountain-valley circulation is related to the differential heating and cooling between narrow canyons and wide basins under radiative forcing, that generate up-slope and up-valley winds during daytime and a reversal at night (e.g. Whiteman, 2000). A direct corollary is a strong seasonality in wind regimes, particularly inland in the Santiago basin, in connection with the latitudinal displacement of the SPH and contrasts in solar radiation between winter and summer affecting the mountain-valley circulation (Lapere et al., 2021b).

Several atmospheric measurement campaigns have been conducted in the vicinity of Santiago aiming to investigate the transport of urban BC into the adjacent Andean valleys, each focusing on a different canyon and/or time period (Cordova et al., 2016; Gramsch et al., 2020; Huneeus et al., 2021). Modeling studies also provided insight on this topic, again focusing on specific periods and/or canyons (Gramsch et al., 2020; Lapere et al., 2021a). Although the aforementioned studies complement each other to refine the knowledge on BC transport near Santiago, a more holistic approach both temporally and spatially, with a regional and climatological relevance is missing, to the best of our knowledge.

It is now well established that particulate matter emitted in Santiago has the ability to reach high-elevation areas in the nearby Andes both in summer and winter periods, with transport time of a few hours (Gramsch et al., 2020; Huneeus et al., 2021). Nevertheless, the aforementioned works focus on atmospheric concentrations but do not investigate deposition on snow or ice covered terrain, which is ultimately the issue at stake. Rowe et al. (2019) and Cordero et al. (2022) measured wintertime BC deposition rates in the high Andes but could not conclude on source attribution. Symmetrically, Lapere et al. (2021a) used a model-based approach enabling to trace back the origin of deposited BC on the Andean cryosphere but only for winter conditions and over a limited area.

The objective of this work is to provide a more complete picture in terms of space, time and drivers of urban BC export towards the Andes Cordillera. We use chemistry-transport modeling with WRF-CHIMERE to compute BC deposition rates





on glaciers of the central Chilean Andes for a given year, and highlight differences between summertime and wintertime. Then, based on local observations and reanalysis data, the meteorological processes driving these differences are identified and

extended on a climatological time scale. Finally, future trends in wind speeds in Santiago area are extracted from CORDEX-CORE simulations under RCP2.6 and RCP8.5 scenarios. BC emission rates associated with these two pathways are also considered, to assess a possible evolution in the seasonal asymmetry in BC transport from Santiago to the Andes.

## 2    Data and methods

### 2.1    Modeling setup

The chemistry-transport simulations performed in this work use the same modeling setup as the one adopted in Lapere et al. (2021b) (Table A1). The same set of simulations is used, despite a different perspective. Hereafter are presented only the most important features of the model. For full details and validation of meteorology and pollutants concentrations, the interested reader is invited to refer to Lapere et al. (2021b). Simulations are performed with the CHIMERE chemistry-transport model (Mailler et al., 2017), forced by meteorological fields simulated with the Weather Research and Forecasting (WRF) mesoscale

numerical weather model from the US National Center for Atmospheric Research (Skamarock et al., 2008). Anthropogenic emissions only are considered for BC (i.e. wildfire events, which can be a large BC source, are ignored), and are adapted from the HTAP V2 inventory (Janssens-Maenhout et al., 2015) using the methodology from Menut et al. (2013). The simulation domain as shown in Figure 1 comprises the central zone of Chile, at the 5 km spatial resolution. A set of four simulations is performed: one month in January 2015 with and without emissions from Santiago city (as defined by the black boxes in Figure 1), and one month in July 2015 considering the same sensitivity analysis. Based on this set of simulations, typical

deposition fluxes of BC, including the contribution of emissions from Santiago, for summertime and wintertime can be derived.

The aerosol dry deposition scheme used in CHIMERE for this work is based on Zhang et al. (2001), that relies on a resistance analogy where the deposition velocity $v_d$ can be written as in Eq. 1.

$$v_d = v_s + \frac{1}{r_a + r_s} \tag{1}$$

where $v_s$ is the settling velocity related to the particle mass upon which gravity acts to bring it back to ground level, $r_a$ is the aerodynamic resistance that accounts for turbulence and surface friction and roughness, and $r_s$ the surface resistance including Brownian diffusion, impaction and interception terms. More details and formulas for the different components of Eq. 1 can be found in the CHIMERE model documentation (CHIMERE, 2021).

Another dry deposition scheme is available in CHIMERE based on Wesely (1989). The compared performance of Zhang

et al. (2001) and Wesely (1989) deposition schemes in CHIMERE was assessed over the Arabian Peninsula in Beegum et al. (2020), revealing that Zhang et al. (2001) matched observations better for this region, despite a seasonal variability in performance. The choice to use only one scheme rather than performing a sensitivity analysis with both schemes is discussed in Section 4.





The accuracy of dry deposition fluxes relies on a proper representation of surface properties such as slope, albedo or rough-
ness. The areas of interest are located in steep terrain, with large sub-grid scale variability in terms of these three properties.
We acknowledge that biases may thus arise in the model, but the nonexistence of measurements of such parameters in the re-
gion prevents validation, although their magnitudes appear to be realistic. The roughness length of glaciers in our simulations
is taken around a few millimeters, consistent with usual magnitudes for ice and snow covered ground (e.g. Fitzpatrick et al.,
2019). The end-of-summer albedo of glaciers of the central Chilean Andes is between 0.2 and 0.6 (Shaw et al., 2021), consis-
tent with the mean 0.33 in our WRF-CHIMERE runs. In wintertime this average reasonably increases to 0.72 due to additional
snow cover, consistent with Malmros et al. (2018). Finally, the slope of the largest glaciers in central Chile is comprised be-
tween 15 and 25° (Shaw et al., 2021), which is likely reasonably captured by the large subgrid-scale orography variance in
WRF over the Andes. Size distribution of aerosols also influence deposition fluxes. In this study, aerosols are distributed into
10 size bins with mass median diameters (10 nm, 30 nm, 70 nm, 160 nm, 0.350 nm, 760 nm, 1.7 µm, 3.5 µm, 7 µm, 20 µm),
following a log-normal distribution. Foret et al. (2006) show that including more than 10 bins would not result in a significant
gain in accuracy.

Regarding wet deposition, CHIMERE distinguishes between below-cloud scavenging by rain and by snow. The former uses
the raindrop size distribution from Willis and Tattelman (1989) to compute the impaction of particles by falling droplets. The
latter is based on the parameterization from Wang et al. (2014). The wet deposition process is highly dependent on the proper
representation of precipitation in WRF over complex terrain. This is known to have relatively poor performance, although at
sufficiently high resolution it is possible to capture the main characteristics of seasonal precipitation over the central Andes
(Schumacher et al., 2020).

## 2.2 Observations, reanalysis and future scenario data

Local observational data of wind speed and direction at surface level in the Santiago area is extracted from the Chilean auto-
mated meteorological and air quality monitoring network (Ministerio del Medio Ambiente, 2015), which will be abbreviated
as SINCA in the continuation of this work, and the Chilean weather service stations (Dirección Meteorológica de Chile, 2019),
referred to as DMC thereafter. Wind vertical profiles at hourly frequency in Santiago, for the years 2017 to 2019, are extracted
from the Aircraft Meteorological Data Relay (AMDAR) data available for the Santiago airport, as presented and analyzed in
Muñoz et al. (2022).

In order to analyze the large-scale circulation and vertical wind profiles at climatological time scales, reanalysis data from
ERA5 at the 0.25° resolution for the period 2010–2020 is used (Copernicus Climate Change Service, 2017). In particular,
monthly averaged reanalyses of zonal and meridional winds on pressure levels, together with mean sea-level pressure, cloud
cover and 10 m zonal and meridional winds are incorporated to the analysis.

The analysis of future trends in surface winds in Santiago is based on outputs from the COordinated Regional Climate Down-
scaling EXperiment-COmmon Regional Experiment (CORDEX-CORE) project (Gutowski Jr. et al., 2016; Giorgi et al., 2021)
on the South America domain at the 22 km resolution (SAM-22), for two regional climate models (REMO2015 and RegCM4-
7) driven by three different general circulation models (MOHC-HadGEM2-ES, MPI-M-MPI-ESM and NCC-NorESM1) under





two future scenarios (RCP2.6 and RCP8.5). The variables of interest were downloaded through the Earth System Grid Federation (ESGF) data platform (ESGF, 2014). Wind speeds are considered as the ensemble mean of the six realizations for each
scenario. Wind direction is not available in REMO2015 outputs so that RegCM4-7 data only is used for this variable. These trends in the regional climate are put into perspective with regard to the associated changes in BC emissions in Santiago area. For that, RCP2.6 (van Vuuren et al., 2007) and RCP8.5 (Riahi et al., 2007) emission scenarios are considered. The data was downloaded from the RCP Database (2009).

For ERA5, CORDEX-CORE and RCP emissions data, graphs showing results for Santiago correspond to the nearest grid
point to downtown Santiago (33.5°S, 70.65°W), without interpolation.

### 2.3 Analysis methodology

The methodological approach adopted in this study follows three steps. First, using chemistry-transport modeling for a summer month and a winter month in 2015, an estimate of the deposition flux of BC on the glaciers of the Andes of central Chile is derived, as well as the share attributable to emissions from Santiago city. These time-limited simulations are designed to provide
insight on the seasonality, magnitude and areas most affected by BC deposition. The seasonality in atmospheric circulation is then analyzed and its role in driving the deposition seasonality is discussed. This part relies on multi-annual time series from reanalysis and local observation, in order to put the findings of the chemistry-transport simulations into a climatological perspective. Finally, future projections of this circulation are analyzed using CORDEX-CORE data under RCP scenarios.

Throughout this work, the locations of glaciers are defined based on WGMS and National Snow and Ice Data Center (2012).
The corresponding nearest grid point in the CHIMERE domain is used to compute the corresponding deposition flux on each individual glacier. Rock glaciers are excluded from the analysis as the main concern regarding BC deposition on glaciers is the impact on their albedo. Several localities and canyons in the vicinity of Santiago are considered throughout this work. For more detail on their exact locations, the reader is invited to refer to Figure A1.

## 3 Results

### 3.1 Seasonal differences in BC deposition

Anthropogenic emissions of primary pollutants such as BC feature a pronounced seasonality, as shown in the HTAP inventory (colormap in Figure 1). In particular, this inventory evidences that in summer (January) BC emissions are concentrated in large urban centers such as the Santiago basin (black square in Figure 1) and the Valparaíso (33°S,71.5°W) and Concepción (37°S,73°W) regions. In contrast, wintertime (July) emissions of BC are more evenly scattered across the domain, in relation
with the widespread use of wood burning for residential heating purposes during that season, all along central Chile (e.g. Saide et al., 2016). This additional emission source in wintertime leads to a monthly total flux of BC over the domain 163% more in winter compared to summer, with $123 \, \mathrm{ton \, month^{-1}}$ for January and $324 \, \mathrm{ton \, month^{-1}}$ for July.





As a result of the combination of the seasonality both in emissions and circulation, deposition fluxes of BC greatly differ between January and July 2015 at the scale of central Chile. In particular, the glaciers-averaged BC deposition flux between 30 and 37°S is four times larger in winter, at $634\,\mu g\,m^{-2}\,month^{-1}$, compared to summer with $158\,\mu g\,m^{-2}\,month^{-1}$ (pink labels in Figure 1). In the direct vicinity of Santiago city, as defined by the glaciers located between latitudes 33°S and 34°S (green line in Figure 1), the seasonality is less pronounced with a monthly total deposition flux of $292\,\mu g\,m^{-2}\,month^{-1}$ in summer and $322\,\mu g\,m^{-2}\,month^{-1}$ in winter (green labels in Figure 1).

To the best of our knowledge, two studies report BC loads in snow of the central Andes (Rowe et al., 2019; Cordero et al., 2022), to which our modeling approach is not readily comparable given the limited data available regarding snow and ice properties in the simulation. Indeed, both studies measure the mass content of BC in snow samples in the Andes, integrated over the snow depth but do not provide deposition rates. The modeling setup adopted here does not provide information on snow depth, which makes the computation of BC load impossible. However, a parallel can be established with modeling studies on other high mountain areas in the world. In particular, it is worth noting that the obtained absolute deposition fluxes are considerably larger than what is modeled regionally for the Tibetan Plateau (maximum deposition rate in the monsoon season around $120\,\mu g\,m^{-2}$) (Ji, 2016) and comparable to the results of measurement campaigns on specific glaciers in that same region (between 2.4 and $62.6\,\mu g\,m^{-2}\,day^{-1}$ (Wang et al., 2017) and between 2.6 and $34.6\,mg\,m^{-2}\,year^{-1}$ (Yan et al., 2019) i.e. 72 to $2883\,\mu g\,m^{-2}\,month^{-1}$). The implications of this comparison are discussed in Section 4.

In these absolute totals, the contribution of Santiago emissions is dominant in summertime with 50% of the BC particles deposited coming from the capital city, while it accounts for only 15% in wintertime at the scale of central Chile (pink pie charts in Figure 1). The seasonality in spatial patterns of BC emissions aforementioned, with emissions found mostly in large urban centers in summer but more scattered in winter, accounts for this observation. This seasonal difference is even greater close to the Santiago basin, with 74% of deposition in summer related to Santiago emissions but only 19% in winter between 33°S and 34°S (green pie charts in Figure 1). Summertime total BC emissions are 50% smaller than in wintertime in this area, according to the inventory used in this work. This decrease does not translate into a decrease in deposition. In fact, when considering dry deposition alone, the deposition flux on glaciers in the vicinity of Santiago (33°S–34°S) even increases by 40% in summertime compared to wintertime, despite emissions being halved. In this case, seasonal variations in regional emissions only cannot explain this change. The additional factors explaining these variations are investigated in Section 3.2.

Figure 2 reveals the spatial patterns associated with BC deposition on snow or ice covered areas, for both seasons, at the scale of the simulation domain. For the summer month (Figure 2a and b), most of the deposition is observed in the vicinity of Santiago with totals up to $1\text{-}2\,mg\,m^{-2}$. Further north and south, and on the other side of the Andes, accumulated deposition is below $200\,\mu g\,m^{-2}$, although a larger portion of the region is under the influence of emissions from Santiago, with a contribution above 30% over the majority of the domain. In contrast, for the winter month which features a wider snow cover (Figure 2c and d), a clear north/south demarcation appears. North of approximately 33°S, the accumulated flux remains mostly below $200\,\mu g\,m^{-2}$, whereas south of this latitude, this flux goes up to more than $2\,mg\,m^{-2}$ over the foothills of the Andes, and is found to be above $400\,\mu g\,m^{-2}$ over the entire area, including the Argentinian side of the Andes. Compared to the summer month, the contribution of emissions from Santiago is more localized and mostly below 30%.



In order to further investigate the large difference in the contribution of Santiago emissions to BC deposition on Andean glaciers in the vicinity of the capital city, Figure 3 provides a more detailed picture with glacier-by-glacier simulated BC
deposition rates and relative contribution for that region. In absolute value, deposition rates feature a large spatial heterogeneity in both seasons, with affected areas varying between summer and winter (Figure 3a and b). In summertime, deposition is larger on glaciers closer to Santiago and decreases further away, with a smooth gradient both in the west-east and north-south directions. It reaches a maximum slightly north of the city, along the Olivares river canyon and near the tip of the Mapocho river canyon, at more than $1\,\mathrm{mg\,m^{-2}\,month^{-1}}$ for the month of January (red circles in Figure 3a). In winter, the BC deposition
flux is spatially more erratic (Figure 3b), in line with the previous comment on emissions being scattered across the region. Maximum values are obtained near the south of Santiago, particularly near the tip of the Maipo river canyon, where deposition rates rise up to $1\,\mathrm{mg\,m^{-2}\,month^{-1}}$. Deposition hot spots are also found further south in connection with the presence of other large urban areas along the Andes.

The more heterogeneous picture in wintertime is consistent with the underlying time dynamics of deposition. In summer, the
glaciers-averaged BC deposition time series features a pronounced, smooth diurnal cycle, with a steady daily deposition rate varying between 2 and $7\,\mathrm{\mu g\,m^{-2}\,day^{-1}}$ for the month of January, associated with a relative standard deviation of 17% (Figure A2). In contrast, the same time series for July is more chaotic, featuring no clear variability mode and daily deposition rates between 4 and $68\,\mathrm{\mu g\,m^{-2}\,day^{-1}}$ with a relative standard deviation of 88%. The associated time series of deposition velocity ($v_d$ in Eq. 1) shows similar patterns, and suggests that particulate matter found in air parcels above glaciers have a similar prob-
ability to be deposited in summer and winter, given that $v_d$ has the same magnitude in both seasons ($2.30\,\mathrm{\mu m\,s^{-1}}$ on average in July and $2.18\,\mathrm{\mu m\,s^{-1}}$ in January). The winter time series also includes a wet deposition component that exacerbates its erratic behavior due to sporadic precipitation episodes inducing spikes in deposition. Wet deposition is an important contributor for wintertime BC deposition and accounts for 34% of deposition on glaciers in the simulation domain. This component is absent in summertime due to essentially dry conditions in the region for that season.

Regarding the relative contribution of emissions from Santiago, Figures 3c and d also reveal contrasting patterns between January and July. In the direct vicinity of the capital city (between 33°S and 34°S), the contribution ranges from 50% to 100%, with a northward gradient. North of 33°S and south of 34°S the contribution decreases to around 30%. Contrarily, in July the contribution from Santiago emissions does not exceed 40% and is concentrated in the south, near the Maipo river canyon (southernmost white line connecting Santiago to the Andes in Figure 3d). This finding is consistent with Lapere et al. (2021a)
that evidenced the transport mechanism of pollutants from Santiago to the Andes in wintertime and revealed the critical role of mountain-valley circulation in the Maipo river canyon. In comparison, all of the major canyons seem to significantly channel BC higher up in large amounts in summer (at least 50% in the vicinity of the white lines in Figure 3c). In particular, in the area of the Mapocho river canyon, moderate BC deposition fluxes are found in July, originating mostly from outside Santiago, whereas they are larger in January and mostly attributable to Santiago emissions. This finding is consistent with Gramsch et al.
(2020) that observed more transport of BC inside this canyon in summer than in winter.

Variations in emissions alone do not account for the differences in summertime and wintertime deposition patterns. The mountain-valley circulation being radiatively-driven it is theoretically more intense in summer, thus implying more channeling





of BC into the canyons, which provides a first additional driver for these differences. Nonetheless, this phenomenon does not account for the reversed gradient in the contribution of Santiago emissions (northward in summer, southward in winter, as observed in Figure 3c and d) given that the magnitude of emissions changes between seasons in our modeling experiment, but the location of the largest sources within the Santiago basin remains consistent. Besides, based on the findings of Lapere et al. (2021a), mountain-valley-circulation-driven transport of BC should primarily affect lower elevation glaciers. However, Figure 4a shows that in summertime, the BC deposition flux on glaciers between 30°S and 37°S is not elevation-dependent (orange line in Figure 4a), although it does indeed decrease with altitude in wintertime (blue line in Figure 4a) confirming the dominant role of mountain-valley circulation for that season.

Assuming a simplified linear relationship between emission and deposition fluxes for given meteorological conditions, the gray dashed line in Figure 4a shows the deposition profile that could be observed in summertime under wintertime BC emission levels. In other words, this line corresponds to the summertime vertical profile of deposition flux, multiplied by the ratio between winter and summer BC emissions in Santiago. In that case, a maximum in the deposition rate is obtained near 4000 m elevation, more than twice as large as compared with the lowest glaciers, and significantly greater than in wintertime for glaciers at the same elevation. In parallel, even after multiplication, summertime deposition at low elevation is still largely smaller than in wintertime for the same altitude. BC thus is primarily deposited on low-elevation glaciers in wintertime, while it has the ability to deposit on higher-elevation glaciers in summertime where it finds its maximum deposition flux.

The asymmetry on the vertical profile does not stem from a statistical bias, since many glaciers are found both in altitude zones of high and low deposition rates in both seasons as shown in Figure 4b. Between 30°S and 37°S, glaciers are predominantly found at two main elevations: around 3000 m and around 4500 m (count maxima in Figure 4b). The former, vertically closer to emission sources, feature large deposition fluxes in winter and small fluxes in summer. The latter show similar fluxes in absolute values for both seasons, but an emission-corrected summertime flux larger than in winter. This indicates a possible ability for summertime emissions to reach higher elevations in connection with more favorable atmospheric conditions for the export of urban air masses.

The homogeneity of the summer vertical profile deposition (Figure 4a), and the more homogeneous horizontal spatial distribution in summer than in winter (Figure 3a) suggest that for summer months, larger-scale atmospheric transport phenomena are likely at play compared to winter. This idea is further investigated in Section 3.2.

In conclusion, the chemistry-transport modeling experiment evidences strong differences in the horizontal and vertical patterns in BC deposition between winter and summer, as well as in the underlying time dynamics, although deposition velocities suggest a similar ability for BC to deposit onto glaciers once above them. As a result, the following hypothesis regarding the processes driving the export of urban BC towards the Andes can be derived: in wintertime, erratically occurring mountain-valley-circulation-driven export dominates the deposition flux, while in summertime the mountain-valley circulation amplifies and a synoptic-scale motion superimposes, leading to BC transport higher up and in a different direction than in wintertime, as discussed in Section 3.2. The next sections investigate the validity of these hypotheses, within a climatological framework through the analysis of ERA5 data and local observations of atmospheric circulation.





## 3.2 Role of local and synoptic-scale circulation

Synoptic-scale circulation in the central zone of Chile is driven by the position of the SPH, in conjunction with the passage of migratory anticyclones regularly leading to the formation of coastal lows disrupting circulation for a few days (e.g. Garreaud et al., 2002). In particular, the contours in Figure 5 show that for the period 2010–2020, the SPH is on average located near 35°S,100°W in January and 5° northward in July near 30°S,100°W. As a result, strong southwesterlies along the Chilean coast blow at the latitude of Santiago in January, whereas weaker southerlies are observed in July. This synoptic forcing leads to dominant southerwesterlies inland as far as Santiago (black box in Figure 5) in January but no established regime in July. The summertime synoptic wind direction is thus consistent with the orientation of the Mapocho and Olivares canyons neighboring the glaciers where maximum deposition and contribution from Santiago are observed in the chemistry-transport simulations at that period. This points to synoptic winds playing a key role in the transport of BC from the urban basin towards the Andes.

Simultaneously, cloud cover is mostly nonexistent above Santiago in January with less than 20% cloud cover on average, while the months of July exhibit a 50% cloud cover (colormap in Figure 5). Consequently, solar radiation fully reaches the ground in summertime, resulting in very active mountain-valley circulation. In contrast, surface radiation is lesser and more variable in wintertime, leading to a comparatively weaker and more erratic mountain-valley circulation, and by extension less BC export through this process. Therefore, in addition to synoptic winds, mountain-valley circulation also governs BC transport in summertime and superimposes on large scale motions. Otherwise, the large deposition rates obtained in the chemistry-transport simulations along the Maipo river canyon, which has a NW-SE orientation perpendicular to the synoptic wind direction, would not be observed if only the synoptic scale played a role. This synoptic/local-scale interaction is consistent with Zängl (2009) for instance, that observed an influence of synoptic forcing on valley-wind circulation when the associated direction matches the direction of the considered valley. Mountain-valley circulation is naturally stronger in summertime regardless of cloud cover considerations, due to longer days and smaller solar zenith angles. The difference in cloud cover evidenced here thus exacerbates this phenomenon.

Vertical wind profiles also advocate in favor of a strong contribution of synoptic forcing to wind conditions in Santiago, thereby influencing the export of BC. Figure 6a reveals, based on ERA5 data, that the aforementioned SPH-generated southwesterlies not only dominate near the surface but also throughout the mixing layer up to around 1500 m a.g.l. in summertime at climatological time scales, with average wind speeds between 2 and 3 m s$^{-1}$. Above the mixing layer, a smooth transition towards stronger winds of 6 to 8 m s$^{-1}$ is observed, with a primary northerly component. The picture is different in wintertime: average wind speeds in the mixing layer up to 1000 m a.g.l. do not exceed 1 m s$^{-1}$ and do not feature a dominant direction. Winds aloft however are much stronger, well above 10 m s$^{-1}$, with a similar direction as in summertime, and create a strong wind shear at the interface between the mixing layer and the free troposphere.

Local measurements of wind profiles at the airport of Santiago from the AMDAR program for a shorter period (2017–2019) reveal a consistent picture with ERA5 data, and provide an additional dimension in terms of its evolution during the day. Figure 6b (c, respectively) shows the average daily cycle of wind speed (colormap) and direction (arrows) profile in Santiago, averaged over the DJF (JJA, respectively) season. Although the comparison with ERA5 data is not straightforward, similar





patterns emerge with generally lower wind speeds near the surface and shallower boundary layer in JJA. During nighttime, DJF and JJA both exhibit moderate to no wind near the ground up until 10 h L.T. but in summertime wind speed starts to increase when solar radiation develops, up to more than $15\,\mathrm{m\,s^{-1}}$ in the afternoon, whereas in wintertime, wind speed decreases in daytime compared to nighttime, reaching values between 2 to $4\,\mathrm{m\,s^{-1}}$ on average. For summertime, wind speeds peaking in

the afternoon is critical when it comes to the export of pollutants, given that it is also a moment when the load of pollutants in the urban atmosphere is high due to accumulation of emissions along the day (white line in Figure 6b). This synchronicity signals an important export potential for contaminants, especially given the direction of these afternoon winds which are southwesterlies, i.e. directed towards the Andes cordillera and its glaciers. Concentrations of $PM_{10}$ are comparatively higher throughout the whole day in winter (white line in Figure 6c) but with weaker winds leading to a lesser export.

Consistent with the conclusions drawn from ERA5 data, these local measurements also suggest that in wintertime (JJA), there is no established wind within the boundary layer, which does not exceed around 1000 m on average, thereby strengthening the accumulation and concentration of pollutants (white line in Figure 6c) and preventing their transport. Contrarily, in summertime (DJF) afternoon winds are well established, blowing towards the Andes, and the deeper and more convective boundary layer also leads to a better ventilation of pollutants outside the urban area.

Therefore, advection of BC by southwesterlies consistently takes place throughout the mixing layer in summer, mostly during daytime when the atmospheric load of pollutants is high, and the smooth transition towards free troposphere winds does not exclude vertical transport outside the mixing layer of a fraction of emissions. In comparison, wintertime emissions are contained within a shallower, stable mixing layer, with no homogeneous horizontal motion and lower wind speeds in the afternoon. These observations reinforce the idea that the summertime BC export is governed by synoptic winds and mountain-valley circulation

combined, while only mountain-valley circulation drives the export in wintertime, with no synoptic component. Although the previous analysis is conducted on a single data point, surface wind observations throughout the Santiago basin show features that are consistent with this proposed seasonality.

Indeed, surface wind observations from the SINCA and DMC networks at different sites in the Santiago basin and inside nearby canyons show a good agreement with the proposed rationale for seasonal change in BC export. Figure 7 compares wind

roses in January and July for four sites: Parque O'Higgins (PQH) in the center of the city (Figure 7a), Las Condes (CND) in the northeast near the Mapocho river canyon entrance (Figure 7b), Puente Alto (PTA) in the southeast near the Maipo river canyon entrance (Figure 7c), and San Jose de Guayacan (SJG) inside the meridional branch of the Maipo river canyon (Figure 7d). A map designating the aforementioned locations and associated canyons can be found in Figure A1.

In downtown Santiago (Figure 7a), there is a clear dichotomy between January and July month winds over the last 15

years. January months feature a predominant southwesterly component in wind direction and wind speeds frequently above $3\,\mathrm{m\,s^{-1}}$. This is consistent with the synoptic forcing observed in Figures 5 and 6. Wind speeds are comparatively slower in July, rarely exceeding $2\,\mathrm{m\,s^{-1}}$, and wind directions exhibit a clear southwesterlies/easterlies dipole consistent with mountain-valley circulation patterns of winds blowing towards the Andes during the day (westerlies) and back towards the urban basin during the night (easterlies). For this location however, there is no easterly component in January, meaning that mountain-valley

circulation is inhibited by the strong synoptic winds.





The observation at the PTA site near the Maipo river canyon entrance (Figure 7c) is similar: strong westerlies well above $4\,\mathrm{m\,s^{-1}}$ dominate in January, and weaker winds forming a mountain-valley dipole are recorded in July. Nonetheless, for this location a summertime easterly component indicative of down-valley winds is observed contrary to downtown, thus evidencing the occurrence of mountain-valley exchanges despite the dominance of synoptic westerlies.

The records show a different situation at PTA's counterpart site, CND, near the entrance of the Mapocho river canyon (Figure 7b). The mountain-valley dipole is observed both in January and July, with a shift from more frequent occurrences of southwesterlies in January to more frequent northeasterlies in July. This is consistent with the summertime synoptic forcing being aligned with the axis of the canyon. Nevertheless, both components of the dipole remain significant and wind speeds are in the same range for both seasons. The shift in the dominant wind direction explains the limited deposition flux in absolute and

relative value observed in July compared to January in the area of the Mapocho river canyon (Figure 3). In July, mostly down-valley winds (northeasterlies) are observed with consequent wind speeds, thus preventing intrusion of BC into the canyon for a majority of the time. In January, wind conditions are the other way around thus favoring the transport of BC from Santiago into the canyon.

Unfortunately, wind speed data inside the canyons are available for only one site, SJG, located in the meridional branch

of the Maipo river canyon (Figure 7d) so that a comparison with what occurs in other canyons is not possible. Nonetheless, it is interesting to observe that at the SJG site, northerlies greater than $5\,\mathrm{m\,s^{-1}}$ dominate the summertime winds, still with a significant component from the south but related to moderate wind speeds. In wintertime, northerlies are still the strongest winds in terms of speed but they are less frequent than southerlies that occur the majority of hours, again associated with moderate wind speeds. Given the configuration of the canyon, northerlies are associated with air masses coming from the

urban basin, thus bringing BC deep into the canyon as evidenced in Huneeus et al. (2021). These winds are the counterpart of westerlies at PTA. As a result, Figure 7d accounts very well for the larger contribution of Santiago emissions to BC deposition fluxes in summertime compared to wintertime near the Maipo river canyon observed in Figure 3c and d: the westerly at PTA / northerly at SJG combination is the most favorable for urban BC intrusions into the canyons, and is recorded more frequently in January than in July. Absolute deposition fluxes in that area are still larger in winter than in summer on average, except near

the Mapocho canyon (Figure 3a and b). Advection conditions are more favorable in summer, but emissions are significantly larger in wintertime, thus explaining the larger share from Santiago but generally smaller absolute total in summer.

In summary, chemistry-transport simulations, reanalysis and local observations consistently evidence and explain that (i) summertime conditions are more favorable for the export of urban BC, explaining that despite lower emissions, deposition rates on individual glaciers can be larger than in wintertime, (ii) synoptic winds influence the summertime transport throughout

the mixing layer, combining with mountain-valley circulation dynamics, while only the latter drives wintertime deposition. Synoptic-scale southwesterlies in summer enable the transport of the BC-containing, more convective mixing layer air masses up to higher elevations. In parallel, mountain-valley circulation also favors the transport of BC through the canyon networks, in both summer and winter, although with a different intensity. The next section briefly explores whether and how this summertime/wintertime dichotomy could evolve under climate change and emission scenarios.





## 3.3 A glimpse at future trends

Figures 8a and b show the time series of future wind speed in Santiago under RCP2.6 and RCP8.5 scenarios, as downscaled within the CORDEX-CORE framework (see Section 2.2 for details on the data and models used). In the context of the RCP2.6 scenario, the ensemble mean does not reveal a significant trend (in the sense of a Mann-Kendall test) for either season (gray lines in Figures 8a and b). Similarly, no trend is observed in the wind direction in that scenario (Figure 8c). Alternatively, the RCP8.5 scenario leads to significant trends (at the 99% level) in wind speed in January and July and in wind direction in January, with +0.02 $\mathrm{m\,s^{-1}\,dec^{-1}}$, -0.03 $\mathrm{m\,s^{-1}\,dec^{-1}}$ and -1.38 $^\circ\,\mathrm{dec^{-1}}$, respectively. In other words, under RCP8.5 scenario hypotheses, summertime wind speeds would increase while wintertime wind speeds would decrease in Santiago area. Although the trends have a limited magnitude, based on the findings of Section 3.2 the summertime/wintertime dichotomy in BC export potential could be exacerbated in the future, with even more favorable conditions for export in summer, and even less favorable export conditions in winter, assuming that, at the first order, BC transport is wind-driven. Summertime emissions could thus matter even more in the coming decades when it comes to BC deposition on Chilean glaciers. A corollary is that due to weaker winds, stagnation of polluted air masses over Santiago in wintertime could potentially increase, thereby worsening air quality conditions in the mega-city. The slight trend in wind direction for January months observed in Figure 8c should not influence the areas where deposition occurs. Neither scenario indicate a trend in wind direction for July months (not shown here).

Along with climate, anthropogenic emissions of greenhouse gases and pollutants also evolve in RCP scenarios. Interestingly, the scenario limiting climate change (RCP2.6) is associated with a lesser reduction in BC emissions compared to the business-as-usual scenario (RCP8.5) for Santiago, with almost no emissions as of 2100 in RCP8.5 but still more than 200 $\mathrm{ton\,yr^{-1}}$ in RCP2.6 by the end of the century (Figure 8d). This comparison is reversed at the global scale where BC emissions remain higher in RCP8.5 (not shown here). The reason for the RCP8.5 featuring lesser BC emissions than RCP2.6 is that the latter is a climate-focused scenario where BC emission reductions are mostly a co-product benefiting from policies on greenhouse gas emissions, while the former assumes that in regions of the world with high exposure to atmospheric pollution, the effort is focused on designing stringent air quality policies (van Vuuren et al., 2011). This category of regions comprises Santiago, hence accounting for comparative trends in emissions in both RCP scenarios.

However, RCP emission scenarios do not provide infra-yearly detail. No information is available on whether the seasonal cycle of emissions would change under such scenarios. Nevertheless, the RCP8.5 climate scenario leads to increasing average temperatures in the Santiago area, especially in winter months (e.g. Cabré et al., 2016). In that case, one can assume that the need for residential heating would either slightly decrease or at most remain constant. Since the difference in BC emissions between summer and winter mostly stems from the contribution of this source, it is likely that under RCP8.5 scenario, the seasonality in BC emissions remains similar or marginally decreases.

In conclusion, although BC emissions in Santiago could likely drop in the coming decades regardless of the scenario, under the RCP8.5 trajectory wintertime emissions could decrease closer to summertime emissions, and the more favorable export conditions in summertime would be exacerbated at the same time. Consequently, despite currently larger regional BC



deposition fluxes on glaciers in wintertime, summer could likely become as equally an important season in the future in this respect.

## 4 Discussion

In this work, the year 2015 only was considered for the simulation of BC deposition fluxes. Figure A3 justifies the relevance of that year on longer time scales by showing time series of $PM_{2.5}$ concentrations and wind speeds in downtown Santiago for each January and July months for the years 2011–2021. In those time series, 2015 features average patterns and magnitudes compared to all the other years of the period. Similarly, the choice to represent winter conditions with the month of July and summer conditions with the month of January, for atmospheric composition and meteorological conditions, is justified by Figure A4, showing that the DJF (JJA, respectively) and January (July, respectively) distributions of $PM_{2.5}$ concentration and wind speed in Santiago cannot be distinguished. January and July are therefore typical months for their respective season.

The chemistry-transport simulations dedicated to estimating the deposition flux of BC on Andean glaciers have been performed with the CHIMERE model using a 5 km resolution domain. Given the steep topography of the region, the sub-grid variability in elevation is large. The altitude of the glaciers as represented in the model might thus be too low compared to reality due to the grid-point averaging. This may result in excessive deposition fluxes, especially in wintertime when deposition is found to be altitude-dependent. Over the 845 glaciers found in the simulation domain, the comparison with data from WGMS and National Snow and Ice Data Center (2012) yields a mean bias of -287 m in the model, associated with a normalized root-mean-square error (NRMSE) of 11% and a Pearson correlation coefficient of 0.85. Thus, the glaciers altitudes are reasonably well reproduced by the model and the bias induced by the representation of the topography is not expected to affect the results by a large margin.

Another large source of uncertainty in the simulations stems from the BC emission fluxes. The HTAP inventory is static as of 2010, and with a relatively coarse resolution compared to the processes studied in this work. Simulated $PM_{2.5}$ concentrations could not be extensively validated across the domain except for a few cities where automated air quality monitoring networks are present, and no validation on the BC fraction of $PM_{2.5}$ could be performed either (see Lapere et al. (2021b) for more detail on the validation). As a result, BC deposition fluxes presented in this work are to be understood as indicative of the order of magnitude rather than precise estimates of the actual fluxes.

In addition, deposition is among the most uncertain processes in chemistry-transport models. Consequently, there is a significant margin for error in the deposition fluxes obtained and presented in Section 3.1. However, a comparison of dry deposition rates simulated by the model and measured by Rowe et al. (2019) for one snowy site in the Andes near Santiago in winter results in a good agreement between both (Lapere et al., 2021a). Although this is only for one site due to lack of more available data, this comparison gives confidence in the model's ability to represent deposition in this topographically complex area. Also, the simulated magnitude is well in line with BC loads measured in Cordero et al. (2022) throughout the Southern Andes, providing additional confidence in the model outputs. Conducting multiple simulations, using different deposition parameterizations could help estimate the variability of our results related to this choice. However, Beegum et al. (2020) showed, for instance, that





the compared performance of both deposition schemes available in CHIMERE depends on the season and location considered. Thus, including a second set of simulations with the other deposition scheme would not result in more confident results. Only extensive comparisons with observations, which are not available, could constrain the uncertainty of the model.

It is worth noting that deposition fluxes of BC obtained with CHIMERE are in line with or above observed and modeled fluxes in the Tibetan Plateau. This region however receives a comparatively greater attention from the research community. The deposition fluxes in Ji (2016) over the Tibetan Plateau, lower than what we found for the Chilean Andes, were estimated to correspond to a radiative forcing of 3 to $6\,\mathrm{W\,m^{-2}}$. A quantitative comparison between the two regions is not adapted here because only two months are considered and therefore the cumulative effect of BC deposition cannot be assessed. Also, there is a lack of information on snow properties in the model for the proper evaluation of BC radiative effect in snow and ice.

Nevertheless, qualitatively this might point to possibly larger impacts on glaciers of the central Andes than the $1.4\,\mathrm{W\,m^{-2}}$ first estimated in Rowe et al. (2019). There are uncertainties attached to our results, but the magnitude of the deposition revealed by our simulations suggests that the radiative impact of BC on the central Chilean Andes glaciers might be currently underestimated. The radiative impact of BC on snow and ice, however, strongly depends on several characteristics of the snowpack, which are not available in our model, and might greatly change between the two regions of the world compared

here. Therefore, it would be interesting in future works to couple the chemistry-transport simulations with a snowpack model in order to estimate directly the radiative forcing implied by BC deposition.

Section 3.1 reveals BC deposition fluxes in summertime which are smaller than in wintertime, but of comparable magnitude. Despite a less extended snow cover in summertime, the impact of the presence of BC in snow and ice is greater at that period compared to wintertime. Indeed, solar radiation being more intense, the difference in heat absorption between pristine

and darkened snow is more noticeable during the melting season (Kang et al., 2020). Consequently, not only is the winter-accumulated BC accelerating the melting, but additional summertime fluxes of BC contribute to worsening this effect. As a result, although the topic of black carbon in snow is often investigated in wintertime because of larger emission fluxes and snow extent, this finding suggests that summertime should not be overlooked, and more closely considered when assessing the modification of the radiative budget by BC on snow and ice.

**5 Conclusions**

This work presented comparative estimates of typical BC deposition fluxes on glaciers of the central Chilean Andes for a summer month and winter month, using chemistry-transport modeling. At the regional scale, this flux is found to be four times larger in winter ($634\,\mathrm{\mu g\,m^{-2}\,month^{-1}}$) than in summer ($158\,\mathrm{\mu g\,m^{-2}\,month^{-1}}$), in connection with the use of wood burning for residential heating purposes. Such fluxes are comparable to that observed in glaciers and snow over the Tibetan Plateau.

While deposition is strongly elevation-dependent in winter with glaciers below $3500\,\mathrm{m}$ featuring the largest fluxes, it is vertically less variable in summer although its maximum is found near $4000\,\mathrm{m}$ elevation. The summertime flux is proportionally larger than in winter assuming equal emissions, pointing to more favorable export conditions. This can be critical since glaciers





melting in summertime provides freshwater to valleys down the way. If Andean glaciers have passed peak water, as is possibly the case according to the literature, their faster melting due to BC deposition will deplete the resource even sooner.

Based on a sensitivity analysis, the share of deposition attributable to emissions from the Santiago basin was estimated at 50% in summer and 14% in winter, related to more scattered BC sources across central Chile in winter. In the vicinity of the capital city (33°S to 34°S), the areas primarily affected by deposition vary significantly from one season to the other, with the north of the city most affected in summer and the south most affected in winter. The rationale for this difference comes from divergent transport mechanisms. Reanalysis and local observation data revealed that synoptic southwesterly winds and

mountain-valley circulation combine in summertime to create ventilation conditions conducive to the transport of BC emissions from urban centers in central Chile towards the Andes in a homogeneous manner, reaching even the highest elevations. In contrast, wintertime only features a more moderate mountain-valley-circulation-driven export but no synoptic forcing, which concentrates deposition close to network of canyons.

In summary, large emissions and moderate mountain-valley circulation lead to large but localized BC deposition fluxes in

winter, while lower emissions but suitable export conditions lead to more widespread BC deposition in summer. Air pollution in the region is mostly investigated in wintertime due to larger emissions and concentrations, but our findings imply that when it comes to the feedback of BC pollution on the albedo of glaciers, and in turn on climate and water resources, summertime is also a season of interest and should not be overlooked.

In the future, according to CORDEX-CORE simulations and emission pathways, under RCP8.5 scenario, this dichotomy in

BC deposition where fluxes are larger in winter could possibly fade however. First, wintertime emissions could potentially drop down and become closer to summertime ones. Second, the favorable summertime export conditions would likely become even more favorable. Third, the bad ventilation conditions in wintertime would likely worsen and therefore be more unfavorable for export.

*Code and data availability.*    The CHIMERE model can be found at http://www.lmd.polytechnique.fr/chimere/CW-download.php. Data from
this work are available from the corresponding author upon reasonable request.

*Acknowledgements.*    This research has been supported by the Agence de l'Innovation de Défense (grant No. NETDESA 2018600074). The collaboration between Chilean and French researchers received support from the ECOS-ANID program action No. C19U02. The collaboration leading to this research has received funding from the European Union H2020 program PAPILA (GA 777544). The chemistry-transport simulations used in this work were performed using the high-performance computing resources of TGCC (Trés Grand Centre de Calcul

du CEA) under the allocation GEN10274 provided by GENCI (Grand Équipement National de Calcul Intensif). RL acknowledges Ricardo Muñoz for providing the AMDAR data and for the fruitful discussions.





*Author contributions.* RL: Conceptualization, Data curation, Investigation, Formal analysis, Visualization, Writing - original draft. NH: Formal analysis, Writing - review & editing. SM: Conceptualization, Writing - review & editing. LM: Conceptualization, Writing - review & editing. FC: Formal analysis.

*Competing interests.* The authors declare that they do not have competing interests.


**Figures**

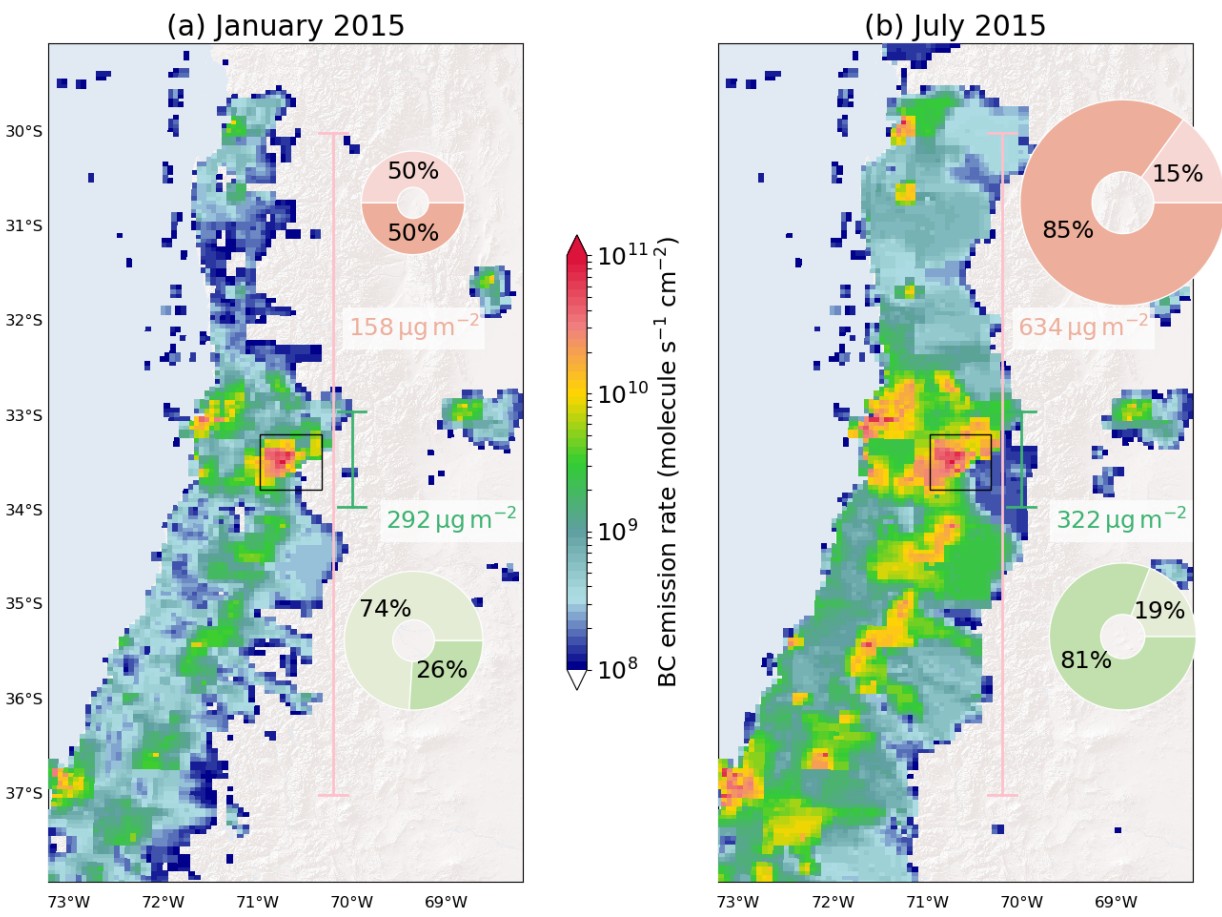

**Figure 1.** Simulated emissions of BC and accumulated deposition on glaciers of the central Chilean Andes in January and July 2015. Color map shows BC emission rates. The black box represents the Santiago city area. Two different zones are considered: 30°S–37°S (pink color code) and 33°S–34°S (green color code). Labels indicate total accumulated BC deposition, pie charts show the share of deposition attributable to emissions from Santiago (lighter color) and other regions (darker color). Pie chart sizes are proportional to the corresponding total deposition. Map background layer source: World Shaded Relief, ©2009 Esri.





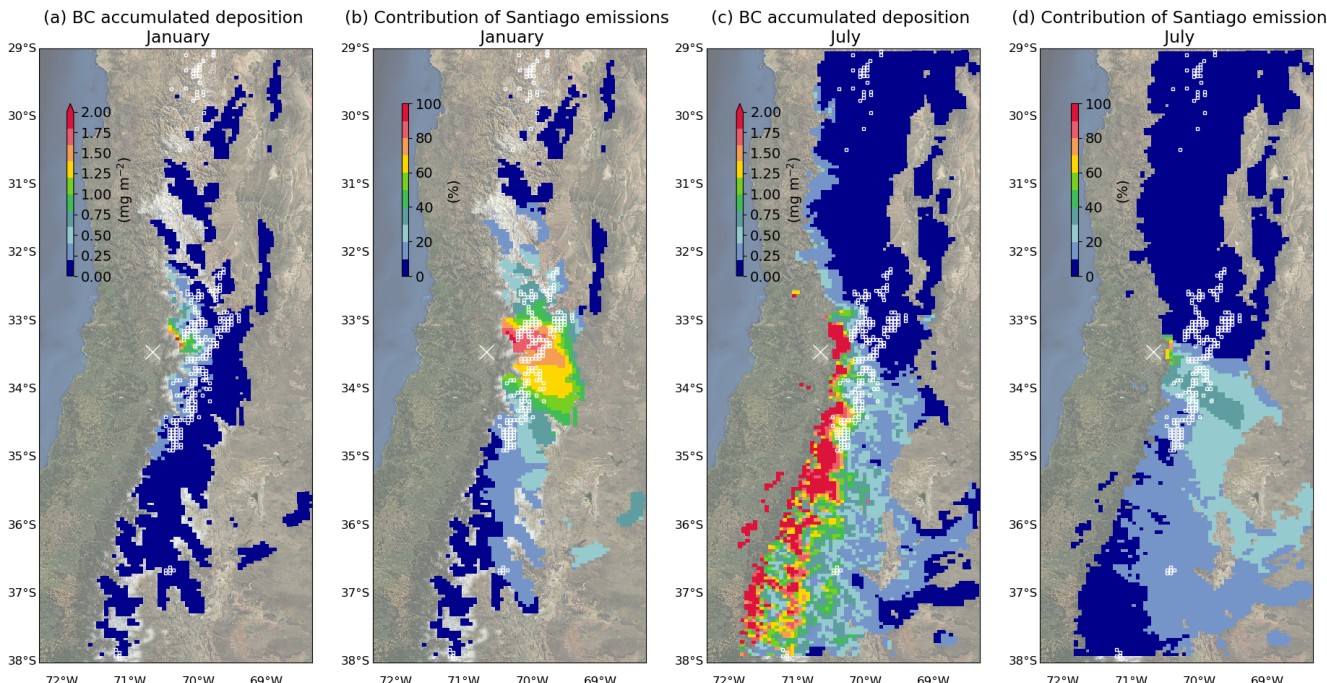

**Figure 2.** (a) Simulated accumulated deposition flux of BC for the month of January. Only grid points that are snow or ice covered are represented. The white cross indicates the location of Santiago. Glacier-covered grid points according to WGMS and National Snow and Ice Data Center (2012) are shown in white squares. (b) Relative contribution of emissions from Santiago area to accumulated BC deposition for January. (c) same as (a) for July. (d) same as (b) for July. Map background layer source: Imagery World 2D, ©2009 Esri.



**Figure 3.** (a) Monthly accumulated BC deposition on glaciers (colored circles) for January 2015. (b) same as (a) for July 2015. (c) Relative contribution of emissions from Santiago to accumulated BC deposition on glaciers for January 2015. (d) same as (c) for July 2015. White lines indicate the main canyons connecting Santiago to the glaciers. Map background layer: Imagery World 2D, ©2009 Esri.





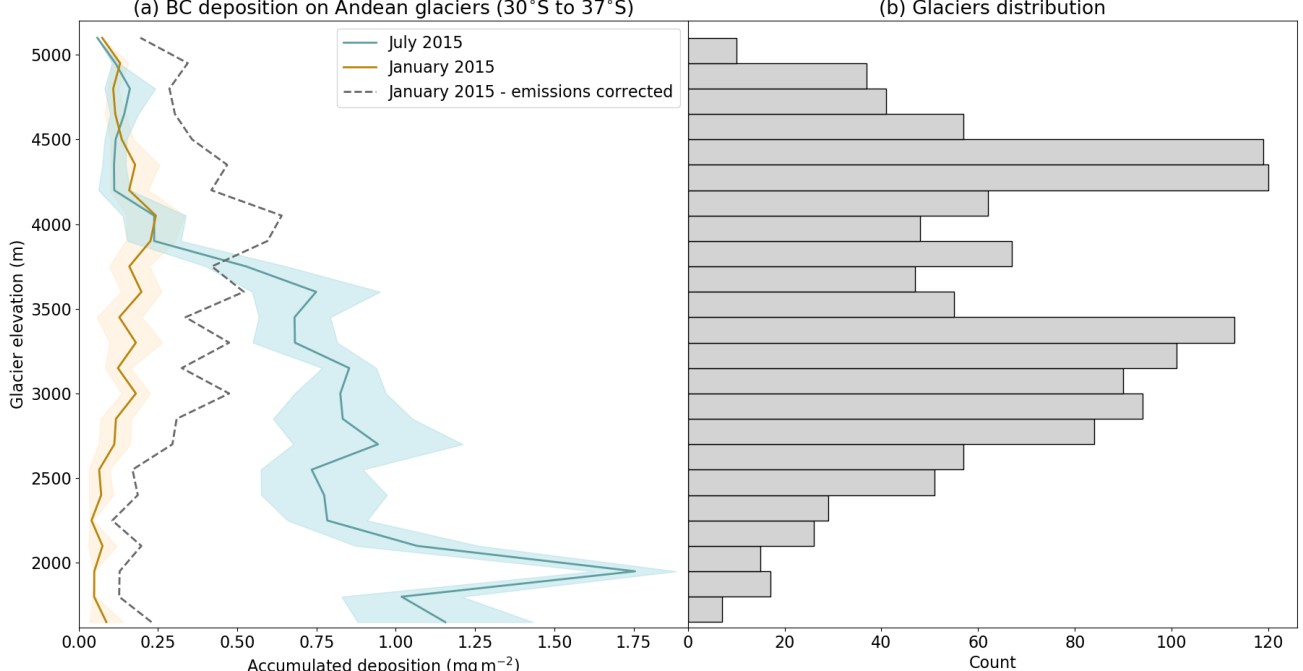

**Figure 4.** (a) BC accumulated deposition on glaciers between 30°S and 37°S depending on their mean elevation. Average for January 2015 (brown line) and July 2015 (blue line). Colored shades show one standard deviation. Dashed grey line is for January 2015 but multiplying the deposition flux by a factor accounting for the difference in emissions compared to July. Glaciers are clustered every 250 m of elevation. (b) Number of glaciers for each model elevation bin (grey bars). In both panels, glacier elevation corresponds to the elevation of the model grid point where the latitude-longitude coordinate of the glacier is found, according to WGMS and National Snow and Ice Data Center (2012).





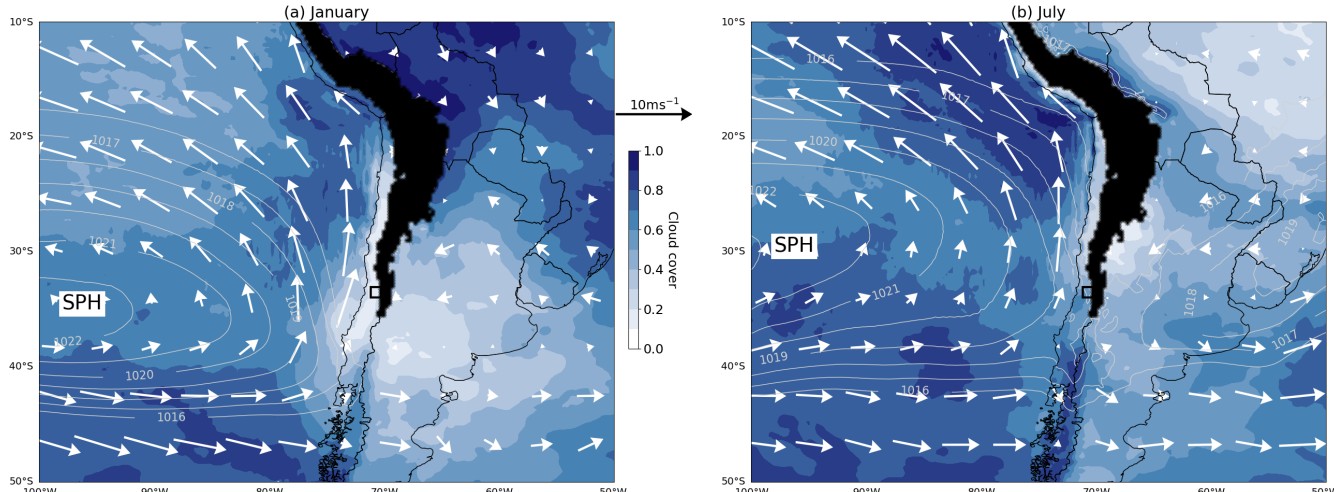

**Figure 5.** Composite synoptic scale circulation for January (a) and July (b) months for the period 2010–2020. Contours indicate mean sea-level pressure (hPa), colormap corresponds to cloud cover, arrows show 10-meter wind speed. The black area represents the Andes where elevation is higher than 2500 m. The black box indicates the location of Santiago. Data from ERA5 single level reanalysis (Copernicus Climate Change Service, 2017).

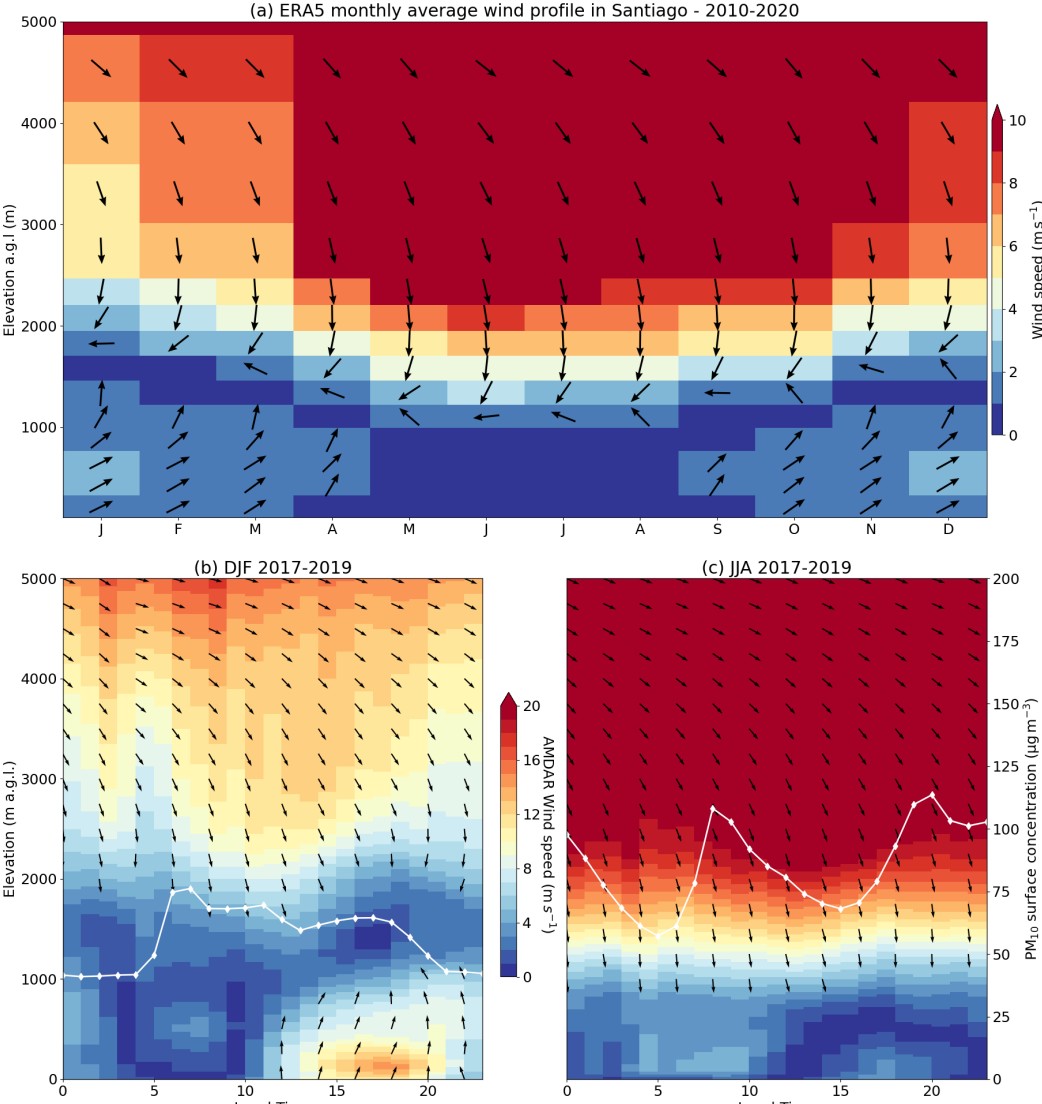

**Figure 6.** (a) Monthly wind vertical profile in Santiago for 2010–2020. Colors indicate horizontal wind speed, arrows indicate the meteorological direction of horizontal winds (i.e. upwards is southerly, rightward is westerly). Directions are displayed for average wind speeds above $1\,\mathrm{m\,s^{-1}}$ only. Average of monthly means from ERA5 pressure-levels reanalysis. (b) Average daily cycle of wind vertical profiles in Santiago from AMDAR for DJF 2017–2019 (colormap) and corresponding $PM_{10}$ surface concentration daily cycle (white line) at PQH station (data from SINCA). (c) same as (b) for JJA.



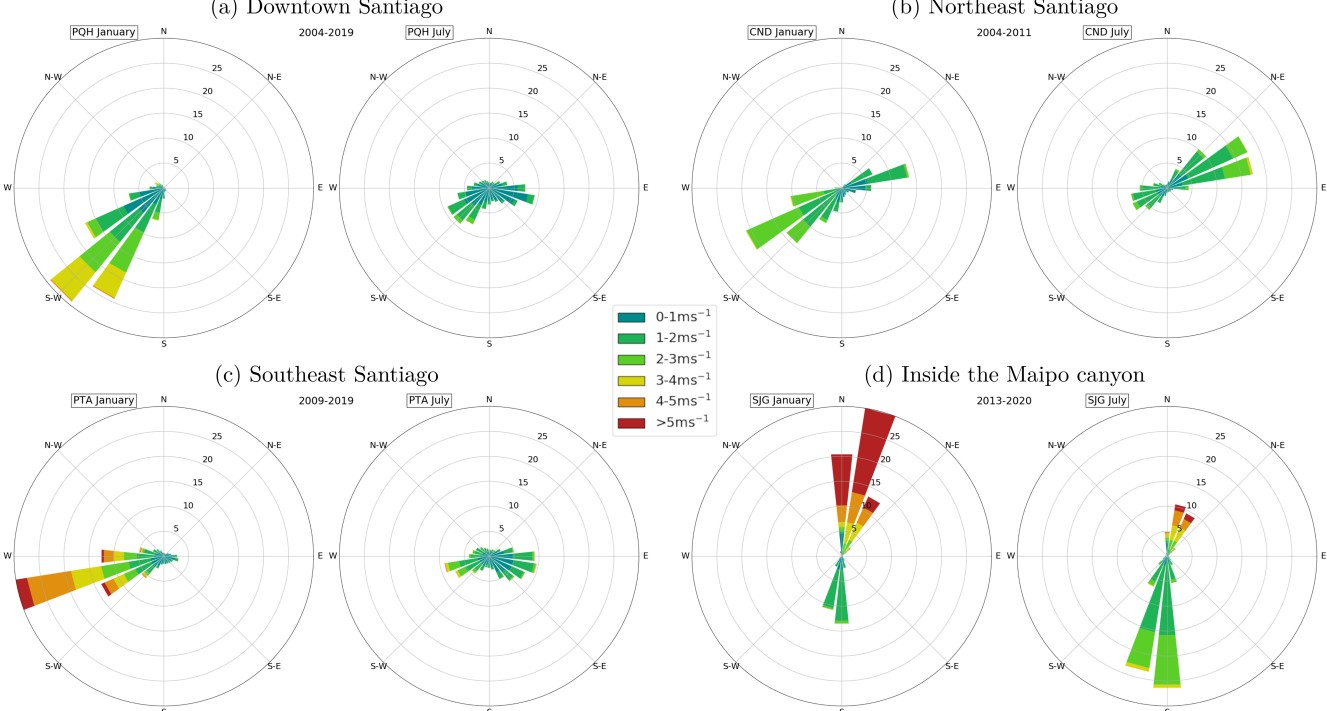

**Figure 7.** Wind speed and direction distribution at (a) Parque O'Higgins (PQH - downtown Santiago), (b) Las Condes (CND - northeast Santiago, near the entrance of the Mapocho canyon), (c) Puente Alto (PTA - Southeast Santiago, near the entrance of the Maipo canyon), (d) San Jose de Guayacan (SJG - deep into the Maipo canyon). Years covered by the underlying time series vary depending on data availability. Wind roses display the distribution of hourly data. Source SINCA for (a), (b), (c), source DMC for (d).





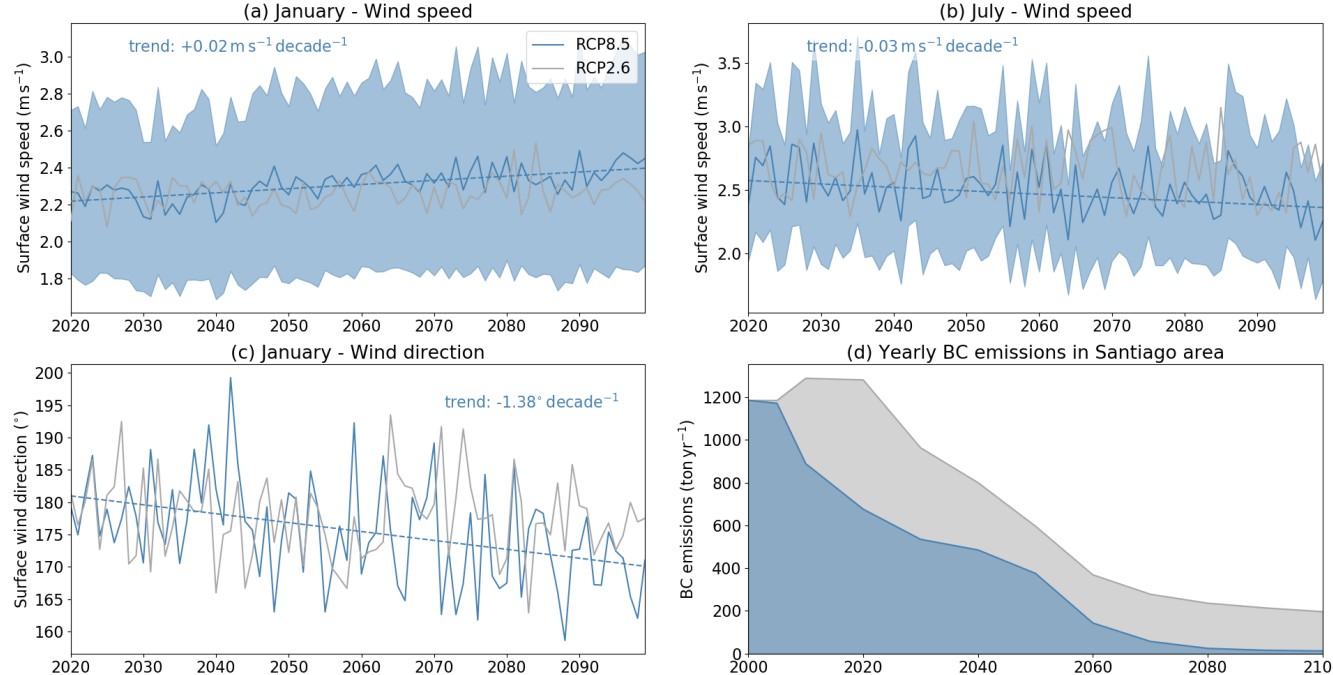

**Figure 8.** (a) Times series of future monthly wind speed in Santiago under scenarios RCP8.5 (blue) and RCP2.6 (gray) for the months of January. Blue shade shows one standard deviation for the RCP8.5 scenario. (b) same as (a) but for the months of July. (c) same as (a) but for wind direction. Ensemble mean of 6 model realizations. Trends significant at the 99% level based on a Mann-Kendall test are shown in dashed line. (d) BC emission scenarios for Santiago under RCP8.5 (blue) and RCP2.6 (gray) scenarios.





# Appendix A

![Figure A1 map]

**Figure A1.** Localities and canyons of interest in the vicinity of Santiago. Measurement stations are designated with white circles. Canyons of interest are evidenced with blue lines. Map background layer source: Imagery World 2D, ©2009 Esri.





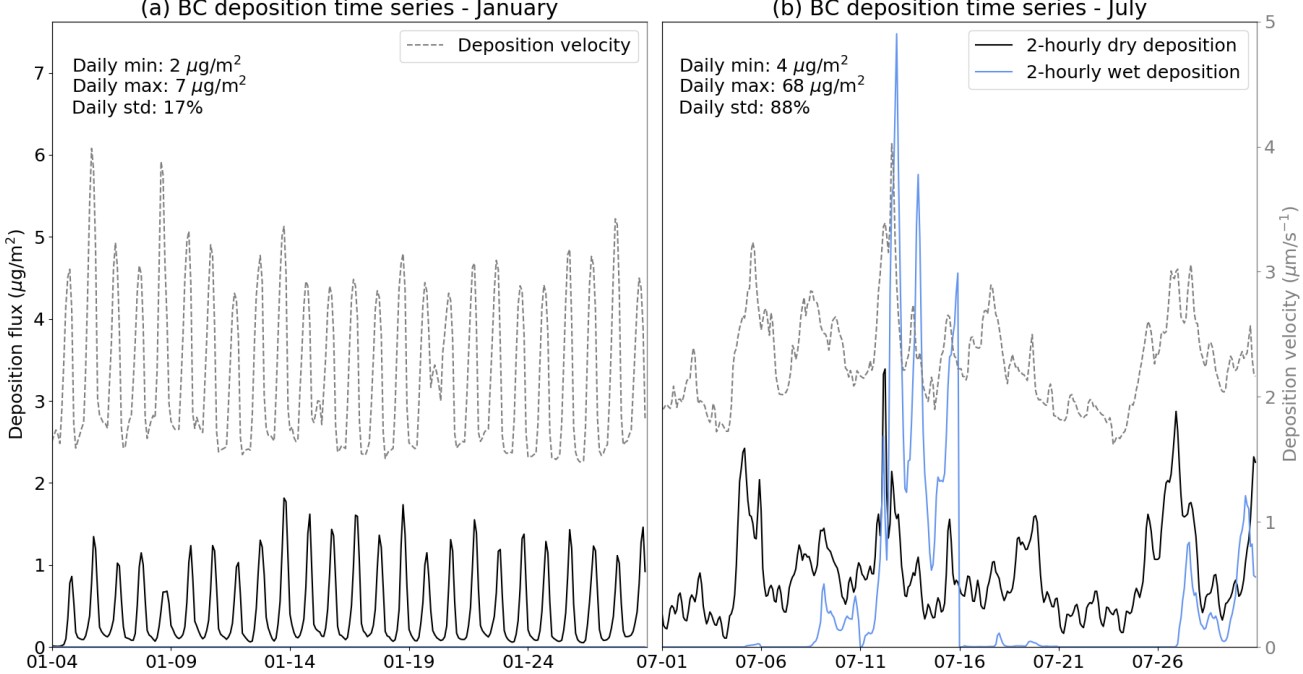

**Figure A2.** Time series of dry (solid black line) and wet (solid blue line) deposition flux and deposition velocity (dashed grey line) in (a) January and (b) July. Average over glacier-containing grid-points within the area considered in Figure 3.

| WRF configuration | | CHIMERE configuration | |
|---|---|---|---|
| | Spatial resolution | 5km | |
| Vertical levels | 60 | Vertical levels | 30 |
| Microphysics | WSM3 | Chemistry | MELCHIOR |
| Boundary and surface layer | MYNN | Gas/Aerosol Partition | ISORROPIA |
| Land surface | Noah LSM | Horizontal Advection | Van Leer |
| Cumulus parameterization | Grell G3 | Vertical Advection | Upwind |
| Longwave radiation | CAM | Boundary Conditions | LMDz-INCA + GOCART |
| Shortwave radiation | Dudhia | | |

**Table A1.** WRF and CHIMERE configurations

# Relevance of year 2015

**Figure A3.** Time series of hourly surface PM$_{2.5}$ concentration (top) and wind speed (bottom) at station PQH in downtown Santiago, in January (left) and July (right). Grey lines are for the years 2011–2021 except 2015. Black line is for the year 2015. Data from SINCA.





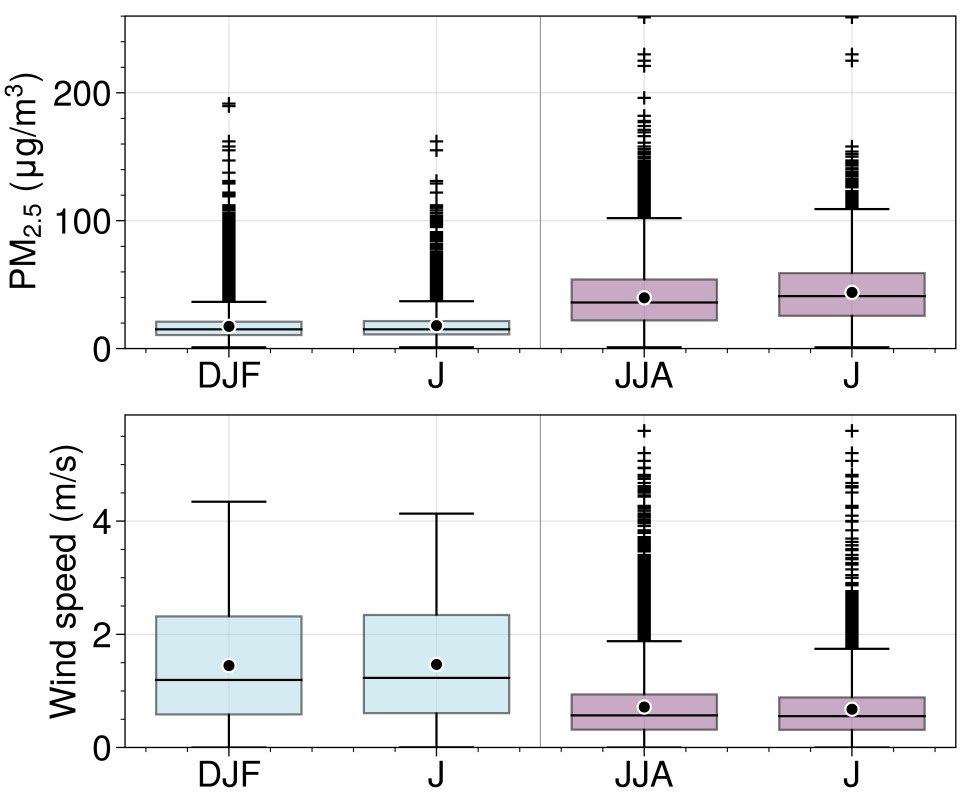

**Figure A4.** Distribution of hourly surface PM$_{2.5}$ concentration (top) and wind speed (bottom) at station PQH in downtown Santiago, in DJF versus January (left) and JJA versus July (right). Data from SINCA for the period 2011–2021.



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
