# Peer review of "Meteorological export and deposition fluxes of Black Carbon on glaciers of the central Chilean Andes"

_Atmospheric Chemistry and Physics, 2022_

## Referee Comment (RC2)

Lapere et al., estimate the seasonal and spatial variability of black carbon deposition fluxes in the Central Andes using chemistry-transport modeling. The drivers of urban BC export towards the Andes Cordillera are also presented. As an important short-lived climate forcer, BC can also reduce the albedo of snow/glaciers when it is deposited on them. Due to the limited observations of BC in atmospheric aerosols and snow/glaciers over Chilean Andes, the simulation of BC deposition is useful for reference for its impact assessment (radiative forcing and snow water equivalent of glacier melting) in this region. However, the significance of your study is a little weak, I strongly recommend the authors to add more emphasis to the MS. But for the section 3, 4 and 5, they are too redundant, try to make them clear and concise, especially for 3.1 and 4. I suggest to move section 4 to section 2, or to the supporting information. Overall, I recommend the current MS to be constructed better and discussed more robust. Thus, the current MS need a major review and a second critical review. The major and specific comments are as followed:

1. You compare the results with the studies on the Tibetan Plateau, however, it is too general. Though, they are glacier regions, but the area, length, volume, and complicated topography, atmospheric circulation, BC sources, and background of BC of the glaciers are different. It is better to add some information, or add the comparison with other glacier regions. There is both atmospheric BC concentrations of and BC depositions over Tibetan Plateau. Does the magnitudes of BC depositions are similar?

2. Rowe et al., 2019 pointed that dust was dominated the albedo reductions in snow rather than BC in northern Chile. Is there possible for you to compare the dust and BC deposition via modelling in your MS?

3. Line 70, I'm puzzled on "Anthropogenic emissions only are considered for BC (i.e., wildfire events, which can be a large BC source, are ignored)", what do you mean?

4. Lines 77-88, You used the aerosol dry deposition scheme from Zhang et al., (2001), according to the better results over the investigation over the Arabian Peninsula. However, you mentioned that "despite a seasonal variability in performance", does

it mean that it is not good for the seasonal variability based on Zhang et al., (2001)? The seasonal variability of BC depositions is the key for your MS. How about to do a comparison based on Wesely (1989) and Zhang et al., (2001)?

5. In current MS, you only compare the influence of BC emissions from and without Santiago city, how about the influence of emissions from a large sale via long-range transport?

6. You focus on the comparison of January and July 2015, but for the meteorological export influence, you use the meteorological parameters during 2010-2020, 2004-2019, 2004-2011, 2009-2019, 2013-2020, they are not coherent.

7. Due to there is no observation data for modelling validation, how about to try to check the MODIS or reanalysis data?

8. You pointed out the future wind speed increase of $+0.02$ m s$^{-1}$, and decrease of $-0.03$ m s$^{-1}$ in January and July, respectively, does the slight variation will really make a big difference under the unclear uncertainties of your current modeling. More importantly, you haven't simulated the radiative fording caused by BC on glacier on Central Andes.

9. L99, "0.350 nm"? It should be "350 nm", right?

---

## Author Comment (AC1)

**Response to the reviewers for ACP-2022-604**

December 21, 2022

**Meteorological export and deposition fluxes of Black Carbon on glaciers of the central Chilean Andes**

Rémy Lapere et al.

Dear Editor and referees,

We acknowledge the Editor and the referees for the time spent to evaluate our work and for their proposed revisions. All of them were answered and included in the new version of the manuscript. Please note that answers are in the boxes after each reviewer's remark. All reviewers' comments were taken into account and are detailed in this letter. Summarizing our answers:

1. Some sections of the manuscript were reorganized following the reviewers' suggestions.

2. Discussion items, including on future trends and modeling choices, were simplified.

3. All figures and tables in Appendix were brought back into the main manuscript and harmonized in terms of the time periods considered.

NB: line and figure numbers are to be understood in reference to the first submitted manuscript, so as to be consistent with the reviewers comments. Changes in the new version imply that these references are no longer relevant in the newly submitted manuscript.

Best regards,
Rémy Lapere
December 21, 2022

**Referee #1**

**General comments**

The work presented by Lapere and co-authors investigates the seasonal and spatial variability of black carbon deposition fluxes in the Central Andes. The topic is not very well represented in current literature; hence the present manuscript is of scientific interest for ACP. Overall, the manuscript is well written, however, the spatial (regional) and temporal (1 year) scale of deposition analysis is not coherent with wind analysis and observations (which goes from continental to local and from decades to hours). From my point of view, this undermines the conclusive message of the manuscript. I do recommend the authors to work on the manuscript for a second round of review. I hope that the following major and minor comments will help the authors in improving the quality of the present manuscript.

**Major comments**

1. INTRODUCTION: The scientific topic of the paper is clearly the identification of sources of BC in snow. However, the forcing mechanisms, and the climatic implications, in the context of the sever draught currently striking Chile, are not sufficiently explained and/or justified in the introduction. If the mass balance of glaciers is dramatically decreasing, modifying the hydrological balance of the region, also the role of BC and its fate will change in the future. What would be the climatic impact of BC in-snow in the current and future context of a short "snow season" and reduced glacier extent? What will be the impact of massive release of BC from the glaciers on the dissolved organic carbon of rivers, lakes and oceans? These questions, which are often disregarded by the BC atmospheric community, should motivate the authors to extend the motivation of their work in the introduction. I might recommend the following literature: https://doi.org/10.1016/j.gloplacha.2022.103837 ; https://doi.org/10.1016/j.earscirev.2017.09.019 ; https://doi.org/10.1038/s41598-019-53284-1 ; https://doi.org/10.1038/nature04141

> **A**: We thank the reviewer for this suggestion to expand the motivations and for providing relevant references. To extend and better frame our motivations in terms of forcing mechanisms and climatic implications, several items have been added in the introduction. Excerpts of these changes are not provided here as they are quite long but appear clearly in the tracked changes version of the manuscript (between lines 25 and 35 of the original manuscript).
>
> Regarding the impact of BC release into water bodies following their deposition onto glaciers and associated melting, the literature seems to indicate that the role of dissolved BC on aquatic ecosystems is not very well known, very dependent on the type of BC, and the mass budget is usually dominated by soil and biomass burning originating BC (see https://doi.org/10.1002/lol2.10076 for example). Both of these sources are not investigated in this work. Therefore, although the question is interesting, we prefer not to discuss this point in the manuscript.

2. SECTION 3.1: This section is particularly long; and, I have the impression that the seasonality and spatial variability of deposition fluxes are repeated multiple times in various subparagraphs and figures. To simplify and shorten the section I recommend the following. 1) Merge Figure 1 and Figure 2, focusing on the seasonality of emissions and deposition fluxes on a larger regional scale, neglecting the direct influence of Santiago emissions (up to me, it is easy to guess that the influence of Santiago emissions will decrease with transport distance). As it is Figure 2 is a mere repletion of Figure 3, with a broader resolution; 2) Focus on the detailed influence of Santiago on the single glaciers as nicely done in Figure 3. Potentially, it might be a good idea to introduce subsections 3.1.1 and 3.1.2.

> **A**: Combining all the reviews, several paragraphs have been removed from Section 3.1. It is now approximately 35 lines shorter compared to the previous version of the manuscript. We believe the flow is now better and introducing additional subsubsections would not improve readability. Figure 2 has been removed, and Figure 1 has been revised to combine information from Figures 1 and 2, as per the reviewer's suggestion.

3. SECTION 3.2: The authors provide many information on wind conditions, potentially a bit more than needed. The deposition fluxes are based on two months of 2015 (July and January), the synoptic atmospheric circulation is based on 10 years of monthly averaged reanalysis data, the wind profile measurements covers the 2017-2019 period with 3 months average and hourly resolution, and the local ground measurements covers a variable spans of years. It is evident that none of the timescales and averaging periods are coherent. Similar discussion could be done for the spatial scale, the manuscript moves from regional scale in Figure 1,2,3,4 to continental scale in Figure 5 and then local to punctual in Figure 6,7. The consequent question is, do all these data, directly, support the analysis of BC regional export and deposition for 2 months of 2015? The authors might think of using solely the local data available for 2015, while synoptic scale might stay as it is.

> **A**: We thank the reviewer for suggesting this improvement. The idea behind this diversity in time periods was to use all available data to make the analysis more robust. We acknowledge this can create confusion and perhaps lose in consistency what is gained in robustness. Accordingly, the data in Figure 6a and in Figure 7 is now for 2015 only. The AMDAR data used in Figure 6b,c is only available for 2017–2019 however so we cannot change the period covered by these panels. Except for these two panels and Figure A3/A4 which describe the relevance of 2015 in a multi-year context, all plots are now coherent and for 2015 only. The text has been adapted accordingly.
>
> Regarding the diversity in spatial scale, it is needed for our analysis, since we explain the transport of BC by the superimposition of local and synoptic-scale circulations.

4. SECTION 3.3: given the wind speed resolution of figure 5, 6, 7 is hard to believe that a change of 0,03 or 0,02 m/s in wind speed would influence export or deposition of BC in 10 years. Similar comment could be done for the 1.4° change in the wind direction for Figure 7. This section is definitely interesting, and should be kept in the final version of manuscript, but the authors should underline the big uncertainty related with this "projection": standard deviation is approximately 10 time larger than starting and ending values (Fig8a-b); precipitation pattern might change, modifying removal mechanisms; etc... Since no future projections for deposition fluxes are ever shown, this section is mostly based on speculation and open guesses.

> **A**: We agree with the reviewer that these trends in wind are indeed small and our conclusions might have been too strong in this regard. To moderate the message this section has been revised. Figure 8 has also been revised, and the discussion on wind direction has been removed.

**Specific comments**

5. L27: Is there an estimation of the forcing caused by dust in snow? Is it comparable to BC?

**A**:

There are estimates comparing the roles of dust and BC. The following sentence was included after line 27 to illustrate the effect of dust on snow compared to BC:

*[Dust emitted by mining activities [Barandun et al., 2022], and mineral dust from the Atacama desert lifted by winds are also found to affect glaciers and snow in northern Chile [Rowe et al., 2019], although the albedo effect for visible wavelengths is scaled down by a factor of about 200 compared to an equivalent mass of BC [Dang et al., 2015].]*

6. L28-35: Here the authors states that Chile is facing an extreme drought, accelerating the melting of glaciers. One can argue that the absence of precipitation and rise of atmospheric temperature are the leading factors causing glacier melting, rather than BC. I suggest the authors explaining more in details the role or the implication for BC budget in snow at line 34-35.

**A**: We acknowledge that the role of precipitation and temperatures changes might supersede the role of BC regarding the retreat of glaciers, although studies suggest LAPs need to be considered whenever looking at trends in snow cover. Accordingly, several items were included in the introduction on this, including:

*[For example, Réveillet et al. [2022] found that BC and dust in snow in the French Alps and the Pyrenees advance the peak melt water run-off by as much as 2 weeks. Similarly, Ménégoz et al. [2013, 2014] found that BC on snow can reduce the snow cover season duration by up to ten days in boreal and temperate regions, and up to eight days in the Himalayas. The evolution of the snow cover with climate change is therefore strongly modulated by the deposition of BC and dust [Réveillet et al., 2022], and BC deposition on Andean glaciers is of great concern although it has received little attention so far compared to other mountain areas in the world [Vuille et al., 2018].]*

*[Glaciers in South America have been evidently retreating in the last 30 years [Rabatel et al., 2013], with impacts on local population and economy [Barnett et al., 2005]. Complex climatic mechanisms are related to this melting in the Andes mountain range, including a combination of atmospheric temperature and precipitation pattern changes [Barnett et al., 2005].]*

7. L51: remove "but"

**A**: Adjusted as suggested.

8. L153: are these total deposition fluxes (dry+wet)? Worth mentioning.

**A**: Yes, the text should mention it. Line 153 now reads: *[...seasonality both in emissions and circulation, total deposition fluxes of BC (dry+wet) greatly differ between...]*

9. F1: colour scale reads "molecules s-1 cm-2". Shouldn't be "particles s-1 cm-2"?

**A**: We thank the reviewer for noticing this issue. The emissions have been converted to $kg/m^2$ in the new version of the figure to make the fluxes easier to make sense of.

10. L159-161: are these studies showing higher BC loading for winter or summer? Are they supporting your deposition seasonal cycle? Without specifying the cycle of BC in concentration in snow, this part of the text does not provide any relevant scientific information and could be easily removed.

> **A:** We agree that it would have been interesting, but the referenced studies do not provide information on seasonality as they only evaluate the load of BC in the snowpack in winter. The intent of this paragraph was to explain why our model is not comparable to such measurements. We agree with the reviewer that this information was not needed and have removed the paragraph accordingly.

11. L163-168: if the "comparison" with Tibetan Plateau is provided in the discussion section, I suggest to minimize the explanation here and simply report that the deposition fluxes of this study are higher or lower than in other regions.

> **A:** We agree. Lines 163–168 have been removed, and the discussion on this comparison with the Tibetan has been moved entirely to the discussion section and simplified.

12. L227-245: I am quite confused by the dashed line in Figure 4a. This is basically scaled with a constant factor using July emission. It should be shown that deposition fluxes are directly proportional to emission intensity, in the supplementary. Overall, this approach is quite questionable, what would happen when scaling deposition fluxes in July using emissions of January? Figure 4b can go in the supplementary. Second comment, glaciers should also be divided in distance from Santiago (or latitudinal bands), especially in the January period. As shown in Figure 3a, the highest deposition fluxes are in the proximity of Santiago in January, at what altitude are those glaciers?

> **A:** We agree that this may have been confusing. Regarding the first part of the comment, the intent of the scaling was to show that for given emissions, BC particles have the ability to be exported higher up in summer than in winter. This is not very visible without re-scaling the summer values, hence this choice (instead of normalizing the data by its maximum we chose to normalize it by the winter/summer emission ratio). We acknowledge this might bring confusion and have decided to remove the dashed line and adapt the x-scale for the summertime profile instead, which leads to the same conclusion. The text has been adapted accordingly. Also, following the other reviewers recommendations, all the figures have now been included in the main manuscript and there is now no Appendix or Supplementary. As a result, we prefer to keep the information from Figure 4b in the main manuscript.
>
> Regarding the second part of the comment, as pointed out by the reviewer, the latitudinal dependence of deposition and contribution from Santiago emissions is already shown in Figure 3. Therefore we are not sure it is needed to include additional information on that. Regarding the question about the altitude of glaciers near Santiago, Figure 1 below shows the same data as Figure 4 (Figure 5 in the new version) in the manuscript but limited to the area near Santiago (33–34°S). This figure shows that generally speaking similar observations can be made for glaciers near Santiago as for glaciers between 30–37°S, and that glaciers in this area are mostly found around 4,000–4,500 m elevation. Given that the general conclusions from Figure 4 (Figure 5 in the new version) also apply for Santiago area, we believe it is not necessary to make this distinction in the manuscript.

13. L249-256: repletion of what already said in L246-249. Group these phrases, avoid repetitions. Do not use "in conclusion".

[Figure]

Figure 1: (a) BC accumulated deposition on glaciers between 33°S and 34°S depending on their mean elevation. Average for January 2015 (orange line) and July 2015 (blue line). Colored shades show one standard deviation. Glaciers are clustered every 250 m of elevation. (b) Number of glaciers for each model elevation bin (grey bars). In both panels, glacier elevation corresponds to the elevation of the model grid point where the latitude-longitude coordinate of the glacier is found, according to WGMS and National Snow and Ice Data Center [2012].

> **A**:  Lines 246–248 and 249–256 have been merged as one single paragraph and trimmed down to avoid repeated information, as suggested.

14. L258: what is the pressure or altitude level for wind speed derived from ERA5?

> **A**:  As indicated in Figure 5 caption, wind speed at 10 meter is used here. To make it more clear, this information is now recalled at line 261.

15. F6 Panel b and c are too detailed, compared to the monthly resolution of deposition fluxes, they should be removed together with the discussion in the text

> **A**:  Deposition fluxes are not only shown as monthly totals as in Figures 1–4 but are also provided at a 2-hourly resolution in Figure A2 (now Figure 4). We believe panels b and c in Figure 6 are important to illustrate when pollutants are more likely to be exported during the day, as discussed in lines 287–300, which can be relevant for future measurement campaigns of aerosol transport in the canyons near Santiago.
>
> In addition, the total deposition fluxes shown in the earlier figures can be seen as the cumulative sum of hourly deposition. The advection of BC does not occur through mean winds but at higher frequency. Therefore, it is interesting to look at what happens at the hourly scale, within those monthly totals to understand the underlying dynamics and processes.

16. F4-6: elevation appears to be different or not well labelled (Figure4). Make it consistent.

> **A**:   We thank the reviewer for noticing this inconsistency. In Figure 4 we show glacier elevation a.s.l., in Figure 6 the vertical coordinate is air mass altitude a.g.l. The labels have been adjusted accordingly in Figures 4 and 6 (now Figures 5 and 7).

17. Section 4: the discussion section repeats, mostly, what already discussed in previous section. Considering the length of the paper, I suggest removing the full section. Part of the section can be implemented elsewhere in the manuscript.

> **A**:   Based on the comments from all the reviewers, we have chosen to keep the discussion section but remove the discussion elements from the other sections of the manuscript and merge them all in the discussion.

18. L459: the manuscript does not provide any proof of BC causing faster melting in the Andes. Plus, very little insights are given on the forcing mechanisms of BC in snow. Authors should be more careful on these generic statements.

> **A**:   Thanks to the first comment of the reviewer, the manuscript now states that BC has been shown to cause faster melting in the Andes. That said, we agree that the statement line 458–459 is not directly connected to the results of our work, and therefore it has now been removed.

19. F8: show standard deviation for both simulations or for none.

> **A**:   Standard deviations have been removed from Figure 8 as per the reviewer's comment.

**Referee #2**

**General comments**

Lapere et al., estimate the seasonal and spatial variability of black carbon deposition fluxes in the Central Andes using chemistry-transport modeling. The drivers of urban BC export towards the Andes Cordillera are also presented. As an important short-lived climate forcer, BC can also reduce the albedo of snow/glaciers when it is deposited on them. Due to the limited observations of BC in atmospheric aerosols and snow/glaciers over Chilean Andes, the simulation of BC deposition is useful for reference for its impact assessment (radiative forcing and snow water equivalent of glacier melting) in this region. However, the significance of your study is a little weak, I strongly recommend the authors to add more emphasis to the MS. But for the section 3, 4 and 5, they are too redundant, try to make them clear and concise, especially for 3.1 and 4. I suggest to move section 4 to section 2, or to the supporting information. Overall, I recommend the current MS to be constructed better and discussed more robust. Thus, the current MS need a major review and a second critical review.
* * *
**A**: All the reviewers had comments that implied some reorganizations within the manuscript, sometimes with requests difficult to merge. We did our best to integrate all these comments, including the suggestions listed in the general and specific comments here. Sections and paragraphs have been reorganized to make the messages clearer and more concise. In particular, all discussion items have now been moved to Section 4, making the other sections easier to read, and approximately 35 lines of text have been removed from section 3.

It is not clear why the reviewer said that the significance of the study is weak after acknowledging and justifying in the sentence just before that this is a useful work. We interpret that as meaning that the introduction may not have contained enough elements of context regarding why BC in snow in the Andes is a topic of major interest. As a result, the introduction has been reviewed and improved to better highlight the importance of conducting such a study.
* * *
**Specific comments**

1. You compare the results with the studies on the Tibetan Plateau, however, it is too general. Though, they are glacier regions, but the area, length, volume, and complicated topography, atmospheric circulation, BC sources, and background of BC of the glaciers are different. It is better to add some information, or add the comparison with other glacier regions. There is both atmospheric BC concentrations of and BC depositions over Tibetan Plateau. Does the magnitudes of BC depositions are similar?
* * *
   **A**: We agree with the reviewer that this is not a like-for-like comparison. With this we intended to highlight the idea that given the same order of magnitude of deposition fluxes obtained, the Andes deserve as much attention as the Tibetan Plateau when it comes to BC deposition on snow and ice, which it is not receiving at the moment. We did not mean to perform an actual comparison since, as the reviewer rightfully states, they are very different regions.

   To avoid sending inappropriate messages, we moved all references to the Tibetan Plateau in the discussion section only and removed it from the conclusions. The discussion paragraph on this topic has also been revised to moderate the conclusions from this comparison.
* * *
2. Rowe et al., 2019 pointed that dust was dominated the albedo reductions in snow rather than BC in northern Chile. Is there possible for you to compare the dust and BC deposition via modelling in your MS?

**A**:  This idea is interesting. However, most of the northern Chile sites included in Rowe et al. [2019], where dust is found to be important for albedo, are above 30°S and thus outside of our simulation domain. According to Rowe et al. [2019] the region we are interested in for this work is more dominated by BC. In theory it would be possible to conduct the comparison suggested by the reviewer, and it would be very interesting but (i) it would only be relevant for a simulation domain more to the north of Chile and (ii) the scope of this work is really the study of the export and deposition of anthropogenic light-absorbing particles, i.e. black carbon. The study of dust from the Atacama desert is thus out of scope.

3. Line 70, I'm puzzled on "Anthropogenic emissions only are considered for BC (i.e. wildfire events, which can be a large BC source, are ignored)" , what do you mean?

**A**:  In our modeling experiment, we are only interested in the role of BC from anthropogenic origin such as traffic, residential or industry. As a result, we do not include BC emitted during wildfire events in our simulations. In CHIMERE, emissions from wildfires come from a separate emission inventory (e.g. CAMS GFAS) than anthropogenic emissions (here HTAP) and are processed through a different part of the code. Therefore it is possible to simply not include them. To clarify this, line 70 now reads: *[... (i.e. wildfire events, which can be a large BC source, are not included in the emission inventory) ...]*

4. Lines 77–88, You used the aerosol dry deposition scheme from Zhang et al., (2001), according to the better results over the investigation over the Arabian Peninsula. However, you mentioned that "despite a seasonal variability in performance", does it mean that it is not good for the seasonal variability based on Zhang et al., (2001)? The seasonal variability of BC depositions is the key for your MS. How about to do a comparison based on Wesely (1989) and Zhang et al., (2001)?

**A**:  We acknowledge this paragraph was not clear. It was not intended to justify the choice of one deposition scheme over the other, but rather to illustrate that previous studies comparing both schemes did not clearly conclude on which one performs best, as discussed in Section 4. The reference to Beegum et al. [2020] has been removed from this paragraph. Regarding the suggestion to compare the two available schemes, as discussed in Section 4 we believe no real usable information could be gained from such an approach unless more deposition measurements are available.

5. In current MS, you only compare the influence of BC emissions from and without Santiago city, how about the influence of emissions from a large scale via long-range transport?

**A**:  We acknowledge that pollutants can reach the glaciers of the Andes through long-range transport, such as BC emitted from biomass burning in the Amazon for example [Magalhães et al., 2019]. This is a very important and interesting topic. However, this is out of the scope of the present work which aims at studying the transport and deposition of pollutants at the local/regional scale. In our modeling experiment, the long range transport is, to some extent, included in the boundary conditions, but these remain the same for the sensitivity tests to isolate only the contribution from local sources, which again is really the focus of this work. The study of long range transport would require a different setup with a hemispheric domain, or switching to a global model.

6. You focus on the comparison of January and July 2015, but for the meteorological export influence, you use the meteorological parameters during 2010–2020, 2004–2019, 2004–2011, 2009–2019, 2013–2020, they are not coherent.

> **A**: The idea behind this diversity in time periods was to use all available data to make the analysis more robust. We acknowledge this can create confusion and perhaps lose in consistency what is gained in robustness. Accordingly, all wind data is now for 2015 only, except for the AMDAR data which we only have for the period 2017–2019.

7. Due to there is no observation data for modelling validation, how about to try to check the MODIS or reanalysis data?

> **A**: We thank the reviewer for this suggestion. We explored this possibility to use MODIS and/or reanalysis data to better constrain our modeling experiments. However, MODIS is known not to have good performance over topographically complex regions and snow or ice covered areas [e.g. Levy et al., 2010]. In addition, MODIS would only allow us to validate AOD but would not say anything about deposition, which is ultimately the more uncertain part of our modeling work since a validation of PM concentration was already carried out for these simulations in Lapere et al. [2021b], as recalled at line 67 of the manuscript. When it comes to reanalysis, they have a coarse resolution (CAMS is 0.75°, MERRA2 is 0.5°) that is not suitable for use in topographically complex areas such as the Andes. Also, since there is no observation data available, it cannot be assimilated in these products and would therefore not provide more realistic atmospheric composition in the region. These products do not provide deposition either.

8. You pointed out the future wind speed increase of $+0.02\,\mathrm{m\,s^{-1}}$, and decrease of $-0.03\,\mathrm{m\,s^{-1}}$ in January and July, respectively, does the slight variation will really make a big difference under the unclear uncertainties of your current modeling. More importantly, you haven't simulated the radiative fording caused by BC on glacier on Central Andes.

> **A**: Regarding the first part of the comment, we agree with the reviewer, and the paragraph discussing future wind speeds has been revised. The new version expands more cautiously on the implications of these relatively small trends.
>
> Regarding the second part of the comment, we did not simulate the radiative forcing of BC on glaciers because this estimation depends on many characteristics of the snow and ice where it is deposited, which require a dedicated snow model. WRF-CHIMERE is an atmospheric model and does not simulate interactions with the snowpack. It would be very interesting to couple our deposition fluxes with a snow/ice model in future work, as we discuss in lines 440–441. However, the scope of the present work is limited to atmospheric transport and deposition pathways. Therefore, an accurate estimate of BC radiative forcing on glaciers in the Andes is not obtainable in this work.

9. L99, "0.350 nm"? It should be "350 nm", right?

> **A**: We thank the reviewer for noticing this typo. It has been corrected accordingly.

**Referee #3**

**General comments**

The article: "Meteorological export and deposition fluxes of Black Carbon on glaciers of the central Chilean Andes" analyzes the BC transport and deposition on glaciers of central Andes during a summer and a winter month. The study addresses important aspects related to the BC transport to the Andes cryosphere, increasing our knowledge on this topic, which has vital implications for water management and availability over the study region. I think the manuscript can be accepted for publication if the authors first address some major and minor comments, which are described below.

**Major comments**

1. I see no reason to include figures as a separate Appendix. I suggest changing the figures in the Appendix to the main text to prevent going back and forward between figures in the main text and those in the Appendix, which slightly worsens the reading fluency.

> **A**: As per the reviewer's recommendation, all figures have now been moved into the main manuscript.

2. L85: This paragraph creates a bit of confusion for me when reading the text. This paragraph seems to justify the use of Zhang et al. (2001) dry deposition scheme in this study based on results obtained by Beegum et al. (2020) in the Arabian Peninsula comparing Zhang et al. (2001) and Wesely et al. (1989) schemes. I think this does not justify Zhang et al.'s choice unless you discuss how the Arabian Peninsula is similar to the Andes. On the other hand, section 4 mentions that "Beegum et al. (2020) showed that the performance of both deposition schemes depends on the season and location considered. Thus, it does not matter which dry deposition scheme is used, if there is not a wide data coverage to assess the model, which is the case. Please, modify the paragraph in L85 to avoid this contradiction. Maybe remove the reference to Beegum et al. (2020) in that paragraph and just mention that a discussion of the choice of the dry deposition scheme will be described in Section 4.

> **A**: We acknowledge this paragraph was not clear. It was not intended to justify the choice of one deposition scheme over the other, but rather to illustrate that previous studies comparing both schemes did not clearly conclude on which one performs best, as discussed in Section 4. Following the reviewer's suggestion, the reference to Beegum et al. [2020] has been removed and the paragraph line 85 now reads:
>
> *[Another dry deposition scheme is available in CHIMERE based on Wesely [1989]. The choice to use only one scheme rather than performing a sensitivity analysis with both schemes is discussed in Section 4.]*

3. L260: Authors mention that synoptic-scale circulations in central Chile are driven by the position of the SPH and the passage of migratory "mid-level" anticyclones. However, synoptic-scale circulations are also affected every year by the passage of cold fronts (mainly between May and September) and cut-off lows to a lesser extent. In particular, the passage of cold fronts in fall and winter bring northwest/west winds to the region that may affect the average circulation shown in Fig. 5b. That, together with the usual seasonal variation of the SPH circulation, may result in less intense southwesterly winds in winter. I suggest the authors include this discussion in this part of the text.

> **A**:    We thank the reviewer for these additional elements.  As per the reviewer's comment, the following has been added at line 266 of the manuscript:
>
> *[These average circulation conditions can be perturbed, especially in winter, by the passage of cold fronts and cut-off lows. These perturbations usually bring northwest/west winds to central Chile, along with precipitation [Falvey and Garreaud, 2007] and therefore result in even less intense southwesterly winds in winter.]*

4. Figures 6 and 7 show mean conditions for circulations. I think the authors should complement those results by analyzing how circulations behave during January and July 2015 to add support to these results. For instance, if authors choose a group of glaciers in the Mapocho basin and another group of glaciers in the Maipo basins with large deposition fluxes, would they be associated with a larger percent of wind directions indicative of synoptic-scale + upward mountain-valley circulations in January and a larger percent of wind directions indicative of down-valley circulations in July in agreement with what the average circulations show?

> **A**:    We thank the reviewer for this suggestion.  As a result of this comment, Figure 2 below has been included in the manuscript.  This figure describes the relationship, in our simulations, between the 2-hourly BC deposition flux on glaciers and wind speed and direction in Santiago, for each month, for glaciers located near Santiago (between $33°S$ and $34°S$).
>
> It shows clearly the influence of the synoptic scale in summer with maximum deposition occurring for southwesterly winds. The contribution of the mountain-valley circulation is harder to illustrate, given the diversity of orientations of canyon axes, causing very diverse mountain-valley circulation wind directions.
>
> A paragraph describing this figure has been added to the manuscript after line 351.

[Figure]

Figure 2: Modeled 2-hourly BC deposition flux (colors) averaged on glaciers between $33°S$ and $34°S$ as a function of modeled wind speed (x axis) and wind direction (y axis) in downtown Santiago ($33.5°S,70.65°W$), for January (left) and July (right) 2015. Gray histograms indicate the density distribution of wind speed and direction, on an arbitrary scale.

5. Section 4: Authors may discuss in that section that although the RCP8.5 is the hypothetically most extreme scenario in the use of fossil fuel, the RCP2.6 scenario maybe will be a more likely scenario

by the end of the century, and results from RCP2.6 may receive more attention than those from the RCP8.5 scenario.

> **A**: We appreciate that this might be a interesting discussion, however we argue that it is not obvious, when it comes to air quality management policies, whether RCP2.6 or RCP8.5 is the more likely scenario. As explained in the manuscript at lines 379–382, for large cities with air pollution issues like Santiago, more measures to curb air pollution are taken than climate-oriented policies. This is somewhat similar to what is currently observed with the decontamination plans designed every 5 years in Santiago. Therefore, regardless of their climate likelihood, it is quite unclear which of these two scenarios would be more likely in terms of BC emissions, which is the most important variable for future BC deposition on glaciers. In any case, the results on future scenarios presented here are quite speculative and illustrative, thus we prefer not to expand on which scenario would be more likely in the context of this work.

6. A winter and a summer month of 2015 was used in the study to better understand BC transport and deposition over central Andes. However, the year 2015 is an El Niño year, which involves large-scale perturbed circulations that may have affected the region much differently during that year. Authors should discuss this in the text.

> **A**: We appreciate this idea. As per the reviewer's suggestion, the following discussion has been appended to section 4 after line 402:
>
> *[The El Niño–Southern Oscillation (ENSO) can affect atmospheric circulation in central Chile. In particular, El Niño and La Niña phases are related to above-average and below-average rainfall, respectively, in winter [Montecinos and Aceituno, 2003]. ENSO also modulates wind speeds in coastal central Chile by affecting the position of the southeastern Pacific anticyclone, leading to generally decreased alongshore winds during El Niño and increased during La Niña [Rahn and Garreaud, 2014]. However, this teleconnection between ENSO and atmospheric circulation in central Chile seems to have weakened during the particular context of the 2010–2018 mega drought [Garreaud et al., 2020]. Therefore, it is difficult to discuss whether the recorded onset of a strong El Niño phase around April–May 2015 affects the relevance of the July 2015 results presented here on longer time scales. However, the previous discussion on observed wind speeds and $PM_{2.5}$ concentrations (Figure 3) supports the hypothesis that 2015 conditions were not exceptional with respect to the surrounding years in terms of the transport of aerosols.]*

7. To show that the results obtained this year are not very different from those obtained in the 2011–2021 period, the authors present the time evolution of PM2.5 concentrations and wind speed at PQH station in downtown Santiago. However, using only one Santiago station in this comparison might not be representative of what authors have shown in the Results section, particularly for winter months since the total BC deposition attributable to Santiago emissions accounts for less than 20% of the total BC deposition. I think that authors should show an overall comparison using all the PM2.5 concentration and wind speed data available in the region (30-37°S) to better show that conditions in 2015 are not very different from the other years. At least they should present this comparison for observations from other towns/cities in the region. There are several SINCA stations in the O'Higgins, El Maule, and Bio Bio regions. Authors could compare mean PM2.5 and wind speed distributions at other sites for January and July between 2015 and the other years or compare how anomalous the regional wind speed and PM2.5 concentrations were compared to the other years of the 2011-2021 period. I leave it out to the authors to choose the best way to show this.

**A**:    Following the reviewer's recommendation, Figure A3 has been revised and now includes data from 5 stations located in 5 different regions in central Chile, as described in the caption of Figure 3 below. Figure A4 has also been revised in the same way and has been merged into Figure A3 as one single figure. In addition, the figure has been moved from the Appendix into the main manuscript. The paragraph starting line 396 has been modified accordingly and now reads:

*[In this work, the year 2015 only was considered for the simulation of BC deposition fluxes. Figure 3 illustrates the relevance of choosing January and July 2015 as representative of pollution and transport conditions in summertime and wintertime, respectively. Figure 3a shows the mean daily surface $PM_{2.5}$ concentrations and wind speeds observed at 5 stations throughout central Chile for the period 2012–2018 in January (left) and July (right), and distinguishes between 2015 (blue) and other years (orange). The envelope of wind speed and $PM_{2.5}$ for 2015 is well contained into the envelope of the other years, for both months. Similarly, Figure 3b shows that the $PM_{2.5}$ concentrations and wind speeds in January and July, for the period 2012–2018, are not different from DJF and JJA conditions, respectively. Therefore, January and July 2015 are typical months, representative of summertime and wintertime conditions, respectively.]*

**Minor comments**

8. L70: The WRF model is not only developed by the National Center for Atmospheric Research (NCAR), it has been developed by different institutions. Based on information from its webpage: "it is a collaborative partnership of the National Center for Atmospheric Research (NCAR), the National Oceanic and Atmospheric Administration (represented by the National Centers for Environmental Prediction (NCEP) and the Earth System Research Laboratory), the U.S. Air Force, the Naval Research Laboratory, the University of Oklahoma, and the Federal Aviation Administration (FAA).

**A**:    The reference to WRF being developed by NCAR has been removed to avoid singling them out as the only developers.

9. L70: Please describe how the authors did not take into account wildfire events. Did they use a database of wildfires to remove those dates from the analysis?

**A**:    In CHIMERE, emissions from wildfires come from a separate emission inventory (e.g. CAMS GFAS) than anthropogenic emissions (here HTAP) and are processed through a different part of the code. Therefore it is possible to simply not include them. Line 70 now reads: *[... (i.e. wildfire events, which can be a large BC source, are not included in the emission inventory) ...]*

10. Please expand a bit more on the discussion of simulation in the text. The authors mention to the reader to look for details in Lapere et al. (2021)a, but still, I think a bit more information about simulations should be provided in this manuscript. For instance, what WRF model version was used in this study, and whether there is a reason supporting the use of this configuration of parameterizations for this region. In addition, include Table A1 in section 2.1.

**A**:    Additional information on model setup and parameterization choices has been included at line 70 as follows:

[Figure]

Figure 3: a) Relevance of the year 2015 for wind speed and PM$_{2.5}$ in January (left) and July (right). Daily mean from hourly data from the SINCA network, for 5 stations throughout central Chile (Andacollo, Viña del Mar, Parque O'Higgins, Rancagua and Talca - see Table 2 for their location). Each color shade corresponds to a different station. Data from 2015 (blue) is compared to data for the period 2012–2018 (orange) excluding 2015. b) Relevance of January and July as representative of summertime and wintertime. Same data as a) but orange is now DJF (left) and JJA (right) and blue is January (left) and July (right).

> *[... with the Weather Research and Forecasting (WRF) mesoscale numerical weather model [Ska-marock et al., 2008], version 3.9.1.1. The set of WRF and CHIMERE parameterizations used in this work (Table 1) have been extensively tested for this region of Chile at several spatial resolutions and show good performance in reproducing meteorology and atmospheric composition over the region of Santiago [Mazzeo et al., 2018; Lapere et al., 2020, 2021a,b]. A spin-up period of 15 days is included before the simulation periods considered here.]*
>
> Also, Table A1 has been moved into Section 2.1 as Table 1.

11. Why did the authors prefer to put Fig. A1 in Appendix A, instead of putting it within the text as Fig. 2? I would suggest including it as Fig. 2 in the text since the text can be read more fluidly. Please

see the major comments above.

> **A:** We agree the figure fits better in the main manuscript. Figure A1 has been moved into the main manuscript as Figure 1. See response to comment n°1.

12. L170: I found the following sentence confusing: "In these absolute totals, the contribution of Santiago emissions is dominant in summertime with 50% of the BC particles deposited coming from the capital city, while it accounts for only 15% in wintertime at the scale of central Chile (pink pie charts in Figure 1)." Since I agree that contributions from other parts of central Chile are dominant in wintertime compared to that from Santiago for the whole of central Chile (85% vs 15%), the contribution in summertime is 50% each. Thus, I suggest changing the word "dominant" to "larger".

> **A:** Accordingly with the reviewer's comment, "dominant" has been replaced with "larger" at line 170.

13. L210: In the direct vicinity of the capital city (between 33°S and 34°S), the contribution ranges from 50% to 100%, with a northward gradient in summertime.

> **A:** "in summertime" has been appended to the sentence line 210 as suggested.

14. Please also include the labels Mapocho and Maipo in Figs. 3b-d

> **A:** The requested labels have been included in the panels in Figure 3.

15. In the description of Figure 4, I suggest avoiding saying that the gray line is a summertime corrected profile. That would imply to me that the summertime profile is not accurate or it is biased and it needs to be corrected by the gray profile, which is not the case. If I understood well, you are only theoretically analyzing how the summertime deposition profile change (together with its implication) if using winter instead of summertime emissions. In addition, I would also suggest changing the legend in Fig. 4a from "January 2015 – emissions corrected" to "January 2015 – wintertime emissions".

> **A:** This part with corrected emissions has been removed following another reviewer's comment.

16. L260: The summertime synoptic wind direction is thus consistent with the orientation of the Mapocho and Olivares canyons (Fig. A1 (or Fig. 2 following the above suggestion)) ...

> **A:** A reference to the figure has been included as suggested.

17. L270: Otherwise, the large deposition rates obtained in the chemistry-transport simulations along the Maipo river canyon (Fig. 3a), which has an NW-SE orientation perpendicular to the synoptic wind direction, would not be observed in this month if only the synoptic scale played a role.

> **A**: A reference to the figure has been included as suggested.

18. L280: Above the mixing layer, a smooth transition towards stronger northerly/northwesterly winds of 6 to 8 m s$^{-1}$ is observed.

> **A**: Line 280 has been modified as suggested.

19. L290: Figures 6b and c show the average daily cycle of wind speed (colormap) and direction (arrows) profile in Santiago, averaged over DJF and JJA, respectively.

> **A**: Line 290 has been modified as suggested.

20. L315: The description of the atmospheric weather station data used in this study should be included in the Data and Methodology section. Another reason to move the figures in the Appendix to the main text. I also suggest including the information of the stations used in the study as a Table in that section with the location (lat,lon) of each station and the period of data availability.

> **A**: A table gathering the required information on stations is now included as Table 2 in Section 2.2. We chose not to expand more on this description in the main text given it is already quite long. Only a sentence referring to the new table has been included after line 114.

21. Fig. 6a: Please detail the latitude and longitude of the Era5 grid-point used to create the plot.

> **A**: The required information is now provided in the caption. The same information has also been included for CORDEX in Figure 8 (now Figure 10) caption.

22. L285: Change sentence to: "Figure 6b and c show the average daily cycle of wind speed (colormap) and direction (arrows) profile in Santiago, averaged over DJF and JJA, respectively.

> **A**: See comment n° 19.

23. Fig. 6 caption: Please spell PQH.

> **A**: The caption in Figure 6 has been modified as suggested.

24. L360: Similarly, no trend is observed in wind direction for that scenario in summertime (Figure 8c).

> **A**: The considerations on wind direction have been removed

25. Figure 8 caption: ... (c) same as (a) but for wind direction in **January**.

> **A**:   The considerations on wind direction have been removed

26. L380: Change infra-yearly by intra-yearly

> **A**:   Modified as suggested.

27. L400: I suggest something like: ” ... showing that the DJF and JJA distributions of PM2.5 concentration and wind speed in Santiago cannot be distinguished from those in January and July, respectively.”

> **A**:   This paragraph has been modified (see comment n° 7).

**References**

M. Barandun, C. Bravo, B. Grobety, T. Jenka, L. Fang, K. Naegeli, A. Rivera, S. Cisternas, T. Münster, and M. Schwikowski. Anthropogenic influence on surface changes at the Olivares glaciers; Central Chile. *Sci. Tot. Environ.*, 833:155068, 2022. doi: 10.1016/j.scitotenv.2022.155068.

T. P. Barnett, J. C. Adam, and D. P. Lettenmaier. Potential impacts of a warming climate on water availability in snow-dominated regions. *Nature*, 438:303–309, 2005. doi: 10.1038/nature04141.

S. N. Beegum, A. Tuomiranta, I. Gherboudj, J. Flemming, and H. Ghedira. Simulation of aerosol deposition flux over the Arabian Peninsula with CHIMERE-2017: Sensitivity to different dry deposition schemes. *Atmos. Res.*, 241:104949, 2020. doi: 10.1016/j.atmosres.2020.104949.

C. Dang, R. E. Brandt, and S. G. Warren. Parameterizations for narrowband and broadband albedo of pure snow and snow containing mineral dust and black carbon. *J. Geophys. Res. Atmos.*, 120:5446–5468, 2015. doi: 10.1002/2014JD022646.

Mark Falvey and René Garreaud. Wintertime Precipitation Episodes in Central Chile: Associated Meteorological Conditions and Orographic Influences. *J. Hydrometeorol.*, 8:171–193, 2007. doi: 10.1175/JHM562.1.

R. D. Garreaud, J. P. Boisier, R. Rondanelli, A. Montecinos, H. H. Sepúlveda, and D. Veloso-Aguila. The Central Chile Mega Drought (2010–2018): A climate dynamics perspective. *Int. J. Climatol.*, 40:421–439, 2020. doi: 10.1002/joc.6219.

R. Lapere, L. Menut, S. Mailler, and N. Huneeus. Soccer games and record breaking $PM_{2.5}$ pollution events in Santiago, Chile. *Atmos. Chem. Phys.*, 20:4681–4694, 2020. doi: 10.5194/acp-20-4681-2020.

R. Lapere, S. Mailler, L. Menut, and N. Huneeus. Pathways for wintertime deposition of anthropogenic light-absorbing particles on the Central Andes cryosphere. *Environ. Pollut.*, 272:115901, 2021a. doi: 10.1016/j.envpol.2020.115901.

R. Lapere, L. Menut, S. Mailler, and N. Huneeus. Seasonal variation in atmospheric pollutants transport in central Chile: dynamics and consequences. *Atmos. Chem. Phys.*, 21:6431–6454, 2021b. doi: 10.5194/acp-21-6431-2021.

R. C. Levy, L. A. Remer, R. G. Kleidman, S. Mattoo, C. Ichoku, R. Kahn, and T. F. Eck. Global evaluation of the Collection 5 MODIS dark-target aerosol products over land. *Atmos. Chem. Phys.*, 10:10399–10420, 2010. doi: 10.5194/acp-10-10399-2010.

Newton de Magalhães, Heitor Evangelista, Thomas Condom, Antoine Rabatel, and Patrick Ginot. Amazonian Biomass Burning Enhances Tropical Andean Glaciers Melting. *Sci. Rep.*, 9:16914, 2019. doi: 10.1038/s41598-019-53284-1.

A. Mazzeo, N. Huneeus, C. Ordoñez, A. Orfanoz-Cheuquelaf, L. Menut, S. Mailler, M. Valari, H. Denier van der Gon, L. Gallardo, R. Muñoz, R. Donoso, M. Galleguillos, M. Ossesa, and S. Tolvett. Impact of residential combustion and transport emissions on air pollution in Santiago during winter. *Atmos. Environ.*, 190:195–208, 2018. doi: 10.1016/j.atmosenv.2018.06.043.

M. Ménégoz, G. Krinner, Y. Balkanski, A. Cozic, O. Boucher, and P. Ciais. Boreal and temperate snow cover variations induced by black carbon emissions in the middle of the 21st century. *The Cryosphere*, 7: 537–554, 2013. doi: 10.5194/tc-7-537-2013.

M. Ménégoz, G. Krinner, Y. Balkanski, O. Boucher, A. Cozic, S. Lim, P. Ginot, P. Laj, H. Gallée, P. Wagnon, A. Marinoni, and H. W. Jacobi. Snow cover sensitivity to black carbon deposition in the Himalayas: from atmospheric and ice core measurements to regional climate simulations. *Atmos. Chem. Phys.*, 14:4237–4249, 2014. doi: 10.5194/acp-14-4237-2014.

A. Montecinos and P. Aceituno. Seasonality of the ENSO-Related Rainfall Variability in Central Chile and Associated Circulation Anomalies. *J. Climate*, 16:281–296, 2003. doi: 10.1175/1520-0442(2003)016¡0281:SOTERR¿2.0.CO;2.

A. Rabatel, B. Francou, A. Soruco, J. Gomez, B. Cáceres, J. L. Ceballos, R. Basantes, M. Vuille, J.-E. Sicart, C. Huggel, M. Scheel, Y. Lejeune, Y. Arnaud, M. Collet, T. Condom, G. Consoli, V. Favier, V. Jomelli, R. Galarraga, P. Ginot, L. Maisincho, J. Mendoza, M. Ménégoz, E. Ramirez, P. Ribstein, W. Suarez, M. Villacis, and P. Wagnon. Current state of glaciers in the tropical Andes: a multi-century perspective on glacier evolution and climate change. *The Cryosphere*, 7:81–102, 2013. doi: 10.5194/tc-7-81-2013.

D. A. Rahn and R. D. Garreaud. A synoptic climatology of the near-surface wind along the west coast of South America. *Int. J. Climatol.*, 34:780–792, 2014. doi: 10.1002/joc.3724.

Marion Réveillet, Marie Dumont, Simon Gascoin, Matthieu Lafaysse, Pierre Nabat, Aurélien Ribes, Rafife Nheili, Francois Tuzet, Martin Ménégoz, Samuel Morin, Ghislain Picard, and Paul Ginoux. Black carbon and dust alter the response of mountain snow cover under climate change. *Nat. Commun.*, 13:5279, 2022. doi: 10.1038/s41467-022-32501-y.

P. M. Rowe, R. R. Cordero, S. G. Warren, E. Stewart, S. J. Doherty, A. Pankow, M. Schrempf, G. Casassa, J. Carrasco, J. Pizarro, S. MacDonell, A. Damiani, F. Lambert, R. Rondanelli, N. Huneeus, F. Fernandoy, and S. Neshyba. Black carbon and other light-absorbing impurities in snow in the Chilean Andes. *Sci. Rep.*, 9:4008, 2019. doi: 10.1038/s41598-019-39312-0.

W. C. Skamarock, J. B. Klemp, J. Dudhia, D. O. Gill, D. M. Barker, M. G. Duda, X.-Y. Huang, W. Wang, and J. G. Powers. A Description of the Advanced Research WRF Version 3. *NCAR Technical Note*, 27, 2008.

Mathias Vuille, Mark Carey, Christian Huggel, Wouter Buytaert, Antoine Rabatel, Dean Jacobsen, Alvaro Soruco, Marcos Villacis, Christian Yarleque, Oliver Elison Timm, Thomas Condom, Nadine Salzmann, and Jean-Emmanuel Sicart. Rapid decline of snow and ice in the tropical Andes – Impacts, uncertainties and challenges ahead. *Earth Sci. Rev.*, 176:195–213, 2018. doi: 10.1016/j.earscirev.2017.09.019.

M. L. Wesely. Parameterization of surface resistances to gaseous dry deposition in regional-scale numerical models. *Atmos. Environ.*, 23:1293–1304, 1989. doi: 10.1016/0004-6981(89)90153-4.

WGMS and National Snow and Ice Data Center. World Glacier Inventory, Version 1. Boulder, Colorado USA. NSIDC: National Snow and Ice Data Center, 2012. Last access 1[st] March 2021.